# Asymptotic Convergence of SGD in Non-Convex Problems: A Stopping Times Method with Relaxed Step-size Conditions

## Abstract

Stochastic Gradient Descent (SGD) is widely used in machine learning research. In previous research, the convergence analyses of SGD under vanishing step-size settings typically assumed that the step sizes satisfied the Robbins-Monro conditions, which is to say, the sum of the step sizes was infinite, while the sum of the squares of the step sizes was finite. In practical applications, a wider variety of step sizes is often used, but these may not meet the Robbins-Monro step-size conditions, thus lacking theoretical guarantees of convergence. To bridge the gap between theory and practical application, this paper introduces a novel analytical method—the stopping time method based on probability theory—to explore the asymptotic convergence of SGD under more relaxed step-size conditions. In the non-convex setting, we prove that the almost sure convergence of the sequence of iterates generated by SGD when step sizes satisfy $\sum_{t=1}^{+\infty} \epsilon_t = +\infty$ and $\sum_{t=1}^{+\infty} \epsilon_t^p < +\infty$ for some $p > 2$. Compared to previous works, our analysis eliminates the need to assume global Lipschitz continuity of the loss function, and it also relaxes the requirement of global boundedness of the high-order moments of the stochastic gradient to local boundedness. Additionally, we prove $L_2$ convergence without the need for assuming global boundedness of loss functions or their gradients. The assumptions required for this work are the weakest among studies with the same conclusions, thereby extending the applicability of SGD in various practical scenarios where traditional assumptions may not hold.

## 1 Introduction

Stochastic Gradient Descent (SGD), originally presented in the seminal work Robbins & Monro (1951), stands as one of the most widely used optimization algorithms in the fields of machine learning and deep learning, due to its simplicity and remarkable efficiency in handling large-scale datasets (LeCun et al. (2002); Hinton (2012); Ruder (2016)). Given these attributes, conducting theoretical research, including convergence analyses, on the SGD is important, as it enables more effective application. This includes understanding the optimal conditions for its use, how to adjust parameters to achieve the best results (Jin et al. (2022)), and determining when it outperforms other optimization methods (Zhou et al. (2020)).

The convergence analyses of SGD in vanishing step-size setting are mainly based on the Robbins-Monro step-size[1] conditions (Robbins & Monro (1951)), which require the step-size to satisfy summability conditions: $\sum_{t=1}^{+\infty} \epsilon_t = +\infty$ and $\sum_{t=1}^{+\infty} \epsilon_t^2 < +\infty$. However, in practical use, a greater variety of step-size is commonly chosen, which often fails to meet the Robbins-Monro step-size conditions, thereby not providing the theoretical convergence guarantees. For example, when the step sizes are $\epsilon_n = \mathcal{O}(1/\sqrt{n})$, the theoretical results based on the Robbins-Monro step-size conditions cannot verify its convergence, but it has achieved someempirical successes (Jin et al. (2022)).

The focus of our work in this paper is to explores the convergence of the SGD under more relaxed step-size conditions, in order to bridge the gap between theory and practical application. We summarize the relevant background and our contributions as following.

---

[1] Also known as the learning rate

**Related Works**. The initial proof of SGD trajectory convergence was attributed to the works of Ljung (1977); Ljung et al. (1987), who established convergence results based on the assumption that the trajectories of the iterates are bounded, i.e., $\sup_{t \geq 1} \|\theta_t\| < +\infty$ (a.s.). However, it is impractical to verify this assumption in advance for real-world scenarios. As a result, in the literature on stochastic approximation, the boundedness of SGD trajectories has remained a condition that is often enforced manually, as seen in the works of Benaïm (2006); Borkar & Borkar (2008); Kushner & Yin (1997), and others. Consequently, theoretical findings that depend on this assumption may hold limited applicability in practice. The ODE method of stochastic approximation, as introduced by Benveniste et al. (2012), is utilized in the work of Mertikopoulos et al. (2020) to demonstrate the convergence of SGD trajectories under more relaxed step-size conditions. This paper reveals that the assumption of trajectory boundedness is inherently satisfied under several standard assumptions, including the global uniform boundedness of the gradients of the loss function (i.e., Lipschitz continuity of the loss function) and the global boundedness of the high-order moments of the stochastic gradient. However, many straightforward optimization scenarios, such as those with the squared loss function, do not meet the Lipschitz continuity criterion. Jin et al. (2022) analyzes the convergence of SGD under relaxed Robbins-Monro step-size conditions, specifically when $\sum_{t=1}^{+\infty} \epsilon_t = +\infty$ and $\sum_{t=1}^{+\infty} \epsilon_t^{2+\delta} < +\infty$, where $0 \leq \delta < \frac{1}{2}$. However, in this work, it is assumed that the set of saddle points of the loss function is empty, and the set of stationary points of the loss function consists of at most finitely many connected components. These assumptions are difficult to verify in advance, thus limiting the application of this theory in practical engineering.

**Our Contributions.** This paper introduces a novel analytical method, the stopping time method based on probability theory, to demonstrate the asymptotic convergence (i.e. almost sure convergence and $L_2$ convergence) of SGD under more relaxed step-size conditions (i.e., $\sum_{t=1}^{+\infty} \epsilon_t = +\infty$ and $\sum_{t=1}^{+\infty} \epsilon_t^p < +\infty$ for some $p > 2$).

**1.** We demonstrate the almost sure convergence of the sequence of iterates generated by SGD without directly assuming trajectory boundedness or the global Lipschitz continuity of the loss function, assumptions that are necessary in the work of Mertikopoulos et al. (2020). Our method futher allows us to limit the requirement of the global boundedness of the $p$-th moment of stochastic gradients to a bounded region, which can be made arbitrarily small when $p \leq 3$.

**2.** Furthermore, we establish the $L_2$ convergence of SGD under the same assumptions. It is important to note that when the loss function is Lipschitz continuous, the almost sure convergence of SGD indeed implies $L_2$ convergence. This result can be readily obtained using the **Lebesgue Dominated Convergence Theorem**. However, if the gradient of the loss function is unbounded, almost sure convergence does not guarantee $L_2$ convergence, as outlined in Remark 1. Therefore, proving the $L_2$ convergence of the SGD algorithm in this scenario is still an intriguing problem.

## 2 PRELIMINARIES

**Problem Formulation:** Suppose the model parameters are denoted by $\theta \in \mathbb{R}^d$ the problem of interest is to minimize the loss function $\min_{\theta \in \mathbb{R}^d} f(\theta)$.

---

**Algorithm 1** Stochastic Gradient Descent (SGD)

---

**Require:** Initialize $\theta_1$
1: **for** $t = 1, 2, \ldots, N$ **do**
2:     Compute the stochastic gradient $g_t \leftarrow \nabla f(\theta_t; \xi_t)$
3:     Update the parameter $\theta_{t+1} \leftarrow \theta_t - \epsilon_t \cdot g_t$
4: **end for**

---

The SGD algorithm is shown in Algorithm1, where $\epsilon_t$ denotes the step-size at the $t$-th iteration, which can be constant or vary over time. In the $t$-th iteration, the stochastic gradient of the loss function is denoted as $\nabla f(\theta_t; \xi_t)$, which provides an unbiased estimate of the true gradient of the loss function, denoted as $\nabla f(\theta_t)$, based on the sampled random variables $\{\xi_t\}_{t \geq 1}$, which are mutually

independent. In the subsequent analysis, we denote $\mathscr{F}_t := \sigma(g_1, \ldots, g_t)$ as the $\sigma$-algebra generated by the stochastic gradients up to the $t$-th iteration , with $\mathscr{F}_0 := \{\Omega, \emptyset\}$ and $\mathscr{F}_\infty := \sigma\left(\bigcup_{t \geq 1} \mathscr{F}_t\right)$.

## 2.1 STEP-SIZE CONDITIONS

In our analysis, the sequence of step sizes $\{\epsilon_t\}$ plays a crucial role in the convergence behavior of the SGD. Different from the Robbins-Monro step-size conditions (i.e., $\sum_{t=1}^\infty \epsilon_t = +\infty$ and $\sum_{t=1}^\infty \epsilon_t^2 < +\infty$), our step-size conditions shown in Setting1 are more general.

**Setting 1** (Assumptions on the Step-size)**.** *Let $\{\epsilon_t\}_{n \geq 1}$ be a sequence of positive monotonic mon-increasing real numbers representing the step sizes used in the optimization algorithm. The sequence $\{\epsilon_t\}_{n \geq 1}$ satisfies the following summability conditions:*

$$\sum_{t=1}^{+\infty} \epsilon_t = \infty, \quad \text{and} \quad \sum_{t=1}^{+\infty} \epsilon_t^p < \infty \ \text{(for some } p > 2).$$

These conditions allow for step sizes that do not necessarily satisfy the Robbins-Monro conditions but are still effective in practice. For example, step sizes of the form $\epsilon_t = \mathcal{O}(1/\sqrt{t})$, which achieve near-optimal sample complexity $\mathcal{O}(\ln T/\sqrt{T})$ (where $T$ is the total number of iterations), are included in our step-size conditions despite violating the Robbins-Monro step-size conditions.

By relaxing the stringent requirements of the Robbins-Monro step-size conditions, our analysis accommodates a broader class of step sizes, thereby enhancing the practical relevance of the theoretical results presented in this work.

## 3 ASSUMPTIONS AND RESULTS

In this section, we will present the basic assumptions required for our proofs, as well as our two main theorems.

### 3.1 ASSUMPTIONS

First, we state the assumptions related to the loss function $f : \mathbb{R}^d \to \mathbb{R}$ used in our analysis.

**Assumption 3.1** (Assumptions on the Loss Function)**.** *Let $f : \mathbb{R}^d \to \mathbb{R}$ be a d-times differentiable function (the loss function). We impose the following conditions:*

(a) *$\textbf{Finite Lower Bound:}$ There exists a real number $f^* \in \mathbb{R}$ such that*
$$f(\theta) \geq f^* \quad \text{for all } \theta \in \mathbb{R}^d.$$

(b) *$\textbf{Lipschitz Continuous Gradient:}$ The gradient mapping $\nabla f : \mathbb{R}^d \to \mathbb{R}^d$ is Lipschitz continuous with Lipschitz constant $L > 0$; that is, for all $\theta_1, \theta_2 \in \mathbb{R}^d$,*
$$\|\nabla f(\theta_1) - \nabla f(\theta_2)\| \leq L\|\theta_1 - \theta_2\|.$$

(c) *$\textbf{Corecivity:}$ $\lim_{\|\theta\| \to +\infty} f(\theta) = +\infty$.*

(d) *$\textbf{Boundedness Near Critical Points:}$ There exists two constants $\eta > 0$, $C_\eta > 0$ such that the sublevel set containing points with small gradient norm is bounded above in function value; explicitly,*
$$\left\{\theta \in \mathbb{R}^d \,\middle|\, \|\nabla f(\theta)\| < \eta\right\} \subseteq \left\{\theta \in \mathbb{R}^d \,\middle|\, f(\theta) - f^* \leq C_\eta\right\}.$$

Item (a) and Item (b) are very classical assumption in stochastic optimization and have been widely used in the analysis of SGD (see, e.g., Bottou (2010); Ghadimi & Lan (2013)). These conditions ensure that the loss function is bounded below and that its gradient does not change too rapidly, which are essential properties for establishing convergence.

Unlike Mertikopoulos et al. (2020), we do not require the loss function itself to be Lipschitz continuous, i.e., we do not impose any boundedness on the gradient. The appearance of Item (c) and

Item (d) is mainly due to our step sizes not satisfying the Robbins-Monro conditions, so we need to impose some restrictions on the behavior of the loss function at infinity to prevent the algorithm from diverging to infinity, which are also uesd in Mertikopoulos et al. (2020).

It is worth noting that, while Item (c) and Item (d) together are fully equivalent to the corecivity and non-asymptotic flatness of the loss function as presented in Mertikopoulos et al. (2020), i.e., $\liminf_{\|\theta\| \to +\infty} \|\nabla f(\theta)\| > 0$, in proving results such as that the trajectory of SGD is an asymptotic pseudotrajectory of the corresponding gradient flow (see Appendix A for related definitions), we only require Item (d). That is, as a standalone condition, Item (d) is weaker than the non-asymptotic flatness condition in Mertikopoulos et al. (2020), as it allows for the existence of infinitely distant stationary points with finite function values.

Next, we specify the assumptions related to the stochastic gradient $g_t$ used in the SGD updates.

**Assumption 3.2** (Assumptions on the Stochastic Gradient). *Let $\{\theta_t\}_{t \geq 1} \subset \mathbb{R}^d$ be a sequence of iterates generated by SGD, and let $\{g_t\}_{t \geq 1} \subset \mathbb{R}^d$ be the corresponding stochastic gradients. We impose the following conditions on $g_t$:*

*(a) **Unbiasedness:** For all $t \geq 1$, $\mathbb{E}[g_t \mid \mathscr{F}_{t-1}] = \nabla f(\theta_t)$.*

*(b) **Weak Growth Condition:** There exists a constant $G > 0$ such that for all $t \geq 1$,*
$$\mathbb{E}\left[\|g_t\|^2 \mid \mathscr{F}_{t-1}\right] \leq G\left(\|\nabla f(\theta_t)\|^2 + 1\right).$$

*(c) **Bounded $2p - 2$-th Moment in a Bounded Region:** (when $p > 3$) There exists a constant $C_p > C_\eta$ such that in the region where $f(\theta) - f^* < C_p$, the following holds for all $t \geq 1$:*
$$\mathbb{E}\left[\|g_t\|^{2p-2}\right] \leq M_p^{\frac{2p-2}{p}},$$
*where $p$ is the same constant as in the step-size conditions (Setting 1). This condition states that in a neighborhood where the loss function values are bounded, the $p$-th moment of the stochastic gradient $g_t$ is uniformly bounded.*

*(d) **Bounded $2p - 2$-th Moment Near Critical Points:** (when $2 < p \leq 3$) Alternatively, there exists an arbitrarily small constant $x > 0$ such that whenever $\|\nabla f(\theta_t)\| < x$, the following holds for all $t \geq 1$:*
$$\mathbb{E}\left[\|g_t\|^{2p-2}\right] \leq M_p^{\frac{2p-2}{p}}, \quad (2 < p \leq 3).$$
*This condition states that near critical points (where the true gradient is small), the $p$-th moment of the stochastic gradient $g_t$ is uniformly bounded. That is, when $2 < p \leq 3$, we only need the $p$-th moment to be bounded in an arbitrarily small region.*

Items (a) and (b) are very classical assumptions in the analysis of SGD and stochastic optimization algorithms. The assumption of unbiasedness of the stochastic gradient (Item (a)) is standard and has been widely used in the literature (see, e.g., Bottou (2010); Ghadimi & Lan (2013)). This assumption ensures that, on average, the stochastic gradient points in the direction of the true gradient, which is crucial for the convergence of the algorithm. The weak growth condition (Item (b)) is also common in the analysis of SGD (e.g., Bottou et al. (2018); Nguyen et al. (2018)). This condition controls the variance of the stochastic gradient by relating its second moment to the norm of the true gradient. It prevents the stochastic gradients from having excessive variance, which could otherwise hinder convergence.

We require the higher-order $p$-th moment boundedness of the stochastic gradient because our step sizes do not satisfy the Robbins-Monro conditions. We need the $2p - 2$-th moment of the stochastic gradient to be bounded.

However, directly assuming that the high order moment of the stochastic gradient is globally bounded (as in Mertikopoulos et al. (2020)) is unreasonable in our setting, where even the true gradient $\nabla f(\theta_t)$ can be unbounded. Fortunately, our new analysis method based on stopping time techniques allows us to restrict the $2p - 2$-th moment boundedness condition to a bounded loss function value region. This leads to Item (c), where we only require the $2p - 2$-th moment of the stochastic gradient to be bounded in regions where the loss function values are below a certain threshold.

Moreover, if $p \leq 3$ in our step-size conditions, we can further relax the requirement by restricting the $2p - 2$-th moment boundedness to an arbitrarily small neighborhood around the critical points (as in Item (c')). This means we only need the $2p - 2$-th moment of the stochastic gradient to be bounded when the true gradient $\nabla f(\theta_t)$ is small. This is a significant relaxation.

These assumptions collectively establish the foundational conditions under which we analyze the convergence of the SGD in subsequent sections. Then we give our main results.

### 3.2 MAIN THEOREMS

We now present our main results, establishing the asymptotic convergence of SGD under the assumptions outlined earlier.

**Theorem 3.1** (**Asymptotic Almost Sure Convergence of SGD**). *Let $\{\theta_t\}_{t \geq 1} \subset \mathbb{R}^d$ be the sequence generated by SGD with initial point $\theta_0$. Under Assumption 3.1, and Assumption 3.2 (When $p > 3$, item (c) is required; When $2 < p \leq 3$, item (d) is required), and with the step sizes $\{\epsilon_t\}_{t \geq 1}$ satisfying Setting 1, the following holds:*

$$\lim_{t \to \infty} \|\nabla f(\theta_t)\| = 0 \quad a.s.$$

This theorem shows that the gradients evaluated at the iterates converge to zero almost surely, indicating that the algorithm approaches a critical point of the loss function along almost every trajectory.

**Remark 1.** (*Almost Sure vs. $L_2$ Convergence*) *As stated in the introduction, almost sure convergence does not imply $L_2$ convergence. To illustrate this, consider a sequence of random variables $\{\zeta_t\}_{t \geq 1}$ where $\mathbb{P}(\zeta_t = 0) = 1 - 1/n^2$ and $\mathbb{P}(\zeta_t = n) = 1/n^2$. According to the Borel-Cantelli lemma, we have $\lim_{t \to \infty} \zeta_t = 0$ a.s. However, it can be shown that $\mathbb{E}[|\zeta_t|^2] = 1$ for all $n > 0$.*

**Theorem 3.2** (**Asymptotic $L_2$ Convergence of SGD**). *Let $\{\theta_t\}_{t \geq 1} \subset \mathbb{R}^d$ be the sequence generated by SGD with initial point $\theta_0$. Under the same assumptions as in Theorem 3.1, the following holds:*

$$\lim_{t \to \infty} \mathbb{E}\left[\|\nabla f(\theta_t)\|^2\right] = 0.$$

This result establishes convergence in the mean square sense, showing that the expected squared norm of the gradients approaches zero as the number of iterations increases and indicating that the convergence of gradient norms across different trajectories is uniform in the $L_2$ norm of the random variables.

In summary, these theorems demonstrate that under our relaxed conditions, SGD converges both almost surely and in $L_2$ to points where the gradient vanishes, even when the gradients may be unbounded and the step sizes do not satisfy the traditional Robbins-Monro conditions.

Given the complexity of the complete proof, we provide an outline of the core ideas in the main text using an analytical framework to streamline the exposition. This framework introduces the stopping time techniques essential for establishing the convergence results. For the full detailed proof, the reader is referred to the corresponding section in the appendix.

## 4 ANALYZING FRAMEWORK

In this section, we present our analysis framework for establishing the convergence of SGD under the relaxed conditions described earlier. Our foundational method remains the Ordinary Differential Equation (ODE) method in stochastic approximation. Our main innovation lies in proving the stability required in the ODE method under the conditions of unbounded gradients and non-Robbins-Monro step-size, as well as in establishing asymptotic $L_2$ convergence. These contributions are detailed in Section 4.1, Section 4.2 and Section 4.3. The fundamental concepts and key results related to stochastic approximation are provided in the Appendix A, to which readers are referred for further details. Here, we only present a classical proposition for determining asymptotic pseudotrajectories in stochastic approximation.

**Property 1** (Proposition 4.1 in Benaïm (2006)). *Let $F$ be a continuous globally integrable vector field. Assume that*

**A1** : *F is Lipschitz and bounded on a neighborhood of $\{x_t : n \geq 1\}$.*

**A2** : *For all $T > 0$,*

$$\lim_{t \to +\infty} \Delta(t, T) := \lim_{t \to +\infty} \sup \left\{ \left\| \sum_{i=t}^{k-1} \gamma_{i+1} U_{i+1} \right\| : k = t+1, ..., m(\tau_t + T) \right\} = 0,$$

*where $\tau_t := \sum_{i=1}^{n} \gamma_i$ for $n \geq 1$, and $m(t) := \sup\{k \geq 1, t \geq \tau_k\}$.*

*Then the interpolated process $X$ is an asymptotic pseudotrajectory of the flow $\phi$ induced by $F$.*

The specific meanings of the corresponding symbols in the theorem can be found in the detailed explanations provided in Appendix A.

### 4.1 THE MAIN DIFFICULTY: PROVING THAT THE EXPECTED SUPREMUM OF THE LOSS FUNCTION VALUES IS BOUNDED

We begin by verifying conditions A1 and A2 in Property 1, thereby establishing that the trajectory of SGD constitutes an asymptotic pseudotrajectory of the corresponding gradient flow. Subsequently, leveraging the additional corecivity assumption, alongside Theorem 5.7 and Proposition 6.4 from Benaïm (2006), we derive the asymptotic almost sure convergence. Building upon this result, we invoke the *Lebesgue's Dominated Convergence* theorem to establish asymptotic $L_2$ convergence. Specifically, we identify a function $h \in \mathscr{F}_\infty$ such that $\mathbb{E}[h] < +\infty$ and $\|\nabla f(\theta_t)\|^2 \leq h$ $(\forall t \geq 1)$, which guarantees the desired $L_2$ convergence.

The main challenge of the proof lies in verifying condition A1 and identifying a suitable Lebesgue control function $h$. However, since we are considering corecivity and $L$-smooth loss functions, these two challenges can actually be reduced to one: proving that $\mathbb{E}\left[\sup_{t \geq 1}(f(\theta_t) - f^*)\right] < +\infty$. We present this as a key lemma below.

**Lemma 4.1.** *Let $\{\theta_t\}_{t \geq 1} \subset \mathbb{R}^d$ be the sequence generated by SGD with initial point $\theta_0$. Under Assumption 3.1, excluding Item (c), and Assumption 3.2 (When $p > 3$, item (c) is required; When $2 < p \leq 3$, item (d) is required), and with the step sizes $\{\epsilon_t\}_{t \geq 1}$ satisfying Setting 1, the following inequality holds:*

$$\mathbb{E}\left[\sup_{t \geq 1}(f(\theta_t) - f^*)\right] \leq M < +\infty, \tag{1}$$

*where $M$ is a constant that depends only on the initial point $\theta_1$ and the constants specified in the assumptions.*

It is worth noting that, if we were only concerned with verifying condition A1, we would only need to prove the relatively weaker statement $\sup_{t \geq 1}(f(\theta_t) - f^*) < +\infty$ a.s.. However, this result alone is insufficient to establish $L_2$ convergence.

Next, we will focus on explaining how to prove this lemma.

In the following analysis, we will frequently use a certain quantity. For convenience, we define it here. We refer to

$$\sum_{t=1}^{T} \mathbb{E}\left[\epsilon_t \|\nabla f(\theta_t)\|^2\right]$$

as the **squared gradient variation**.

Furthermore, we define

$$\sum_{t=1}^{T} \mathbb{E}\left[\epsilon_t^m \|\nabla f(\theta_t)\|^2\right]$$

as the **m-order squared gradient variation**.

### 4.2 Transforming the Expectation of the Loss Function Supremum into Squared Gradient Variations

Our approach is to first transform the expectation of the supremum of the loss function into the expectation of the sum of the one-step squared gradient variations, as shown below (informal):

$$\mathbb{E}\left[\sup_{1\le t < T}(f(\theta_t) - f^*)\right] = \mathcal{O}(1) + \mathcal{O}\left(\sum_{t=1}^{T}\mathbb{E}\left[\mathbb{I}_{f(\theta_t)-f^*\ge C_p}\epsilon_t\|\nabla f(\theta_t)\|^2\right]\right).$$

For the case where $2 < p \le 3$, since $C_p$ is not defined in our assumptions, we set $C_p := C_\eta + 1$. The derivation of this equivalence is straightforward and can be found in the appendix, see Appendix C.8 for the detailed steps.

Thus, it suffices to prove that the sum of the squared gradient variations on the right-hand side of the above inequality is bounded.

### 4.3 Our Innovation: Stopping Time Analysis Method

We first present the full boundedness result for the gradient squared variation in the form of a lemma.

**Lemma 4.2.** *Let $\{\theta_t\}_{t\ge 1} \subset \mathbb{R}^d$ be the sequence generated by SGD with initial point $\theta_0$. Under Assumption 3.1, excluding Item (c), and Assumption 3.2 (when $p > 3$, use Item (c); when $2 < p \le 3$, use Item (d)), and with the step sizes $\{\epsilon_t\}_{t\ge 1}$ satisfying Setting 1, the following inequality holds:*

$$\sum_{t=1}^{+\infty}\mathbb{E}\left[\mathbb{I}_{[f(\theta_t)-f^*\ge C_p]}\epsilon_t\|\nabla f(\theta_t)\|^2\right] < C(p, C_\eta, C_p) < +\infty, \tag{2}$$

*where $C(p, C_\eta, C_p)$ is a constant that depends only on $p, C_p, C_\eta$, and the constants specified in the assumptions. Since $C_p$ is not defined in Assumptions 3.2~(d) when $2 < p \le 3$, we define $C_p := C_\eta + 1$ in this case.*

The complete proof of this theorem is quite intricate, and we do not have sufficient space to provide it in full in the main text. Below, we outline some key steps. The detailed proof can be found in Appendix C.6 and C.7.

We need to bound the following sum of squared gradient variations:

$$\sum_{t=1}^{T}\mathbb{E}\left[\mathbb{I}_{f(\theta_t)-f^*\ge C_p}\epsilon_t\|\nabla f(\theta_t)\|^2\right].$$

It is important to note that for the global sum of squared gradient variations,

$$\sum_{t=1}^{T}\mathbb{E}\left[\epsilon_t\|\nabla f(\theta_t)\|^2\right],$$

it is not possible to prove boundedness under the non-Robbins-Monro step-size condition. However, we only need to prove that the local sum of squared gradient variations is bounded when $\theta_t$ is within the event $[f(\theta_t) - f^* \ge C_p]$. To handle this, we introduce stopping times. Specifically, for any $C_\eta < a < b < C_p$, we construct the following sequence of stopping times based on when the loss function $f(\theta_t) - f^*$ first enters the interval defined by $a$ and $b$.

$\tau_1 := \min\{t \ge 1 : f(\theta_t) - f^* > a\}$, $\tau_2 := \min\{t \ge \tau_1 : f(\theta_t) - f^* > b \text{ or } f(\theta_t) - f^* \le a\}$,

$\tau_3 := \min\{t \ge \tau_2 : f(\theta_t) - f^* \le a\}, ...,$

$\tau_{3k-2} := \min\{t \ge \tau_{3k-3} : f(\theta_t) - f^* > a\}$,

$\tau_{3k-1} := \min\{t \ge \tau_{3k-2} : f(\theta_t) - f^* > b \text{ or } f(\theta_t) - f^* \le a\}$,

$\tau_{3k} := \min\{t \ge \tau_{3k-1} : f(\theta_t) - f^* \le a\}$.

Based on certain stopping time techniques, we can establish the following inequality (informal):

$$\mathbb{E}\left[\mathbb{I}^{(i)}\sum_{t=\tau_{3i-1,T}}^{\tau_{3i,T}-1}\epsilon_t^m\|\nabla f(\theta_t)\|^2\right] \le \mathcal{O}\left(\mathbb{E}[\mathbb{I}^{(i)}]\right) + \mathcal{O}\left(\mathbb{E}\left[\mathbb{I}^{(i)}\sum_{t=\tau_{3i-1,T}}^{\tau_{3i,T}-1}\epsilon_t^{m+1}\|g_t\|^2\right]\right) + R_i, \tag{3}$$

In the above equation, the double-subscript stopping time $\tau_{k,T}$ represents the truncated stopping time $\tau_k$ at the finite time $T$, i.e., for any $T > 0$, we define $\tau_{k,T} := \tau_k \wedge T$. The term $\mathbb{I}^{(i)}$ denotes the indicator function for the event $[\tau_{3i-1,T} < \tau_{3i,T}]$. The term $R_i$ represents a negligible remainder, which satisfies $\sum_{t=1}^{+\infty} |R_i| < +\infty$. For the precise form, see Eq. 23 in the appendix.

By summing both sides of Eq. 3 with respect to $i$, we obtain (informal):

$$\sum_{i=1}^{+\infty} \mathbb{E}\left[\mathbb{I}^{(i)} \sum_{t=\tau_{3i-1,T}}^{\tau_{3i,T}-1} \epsilon_t^m \|\nabla f(\theta_t)\|^2\right] \leq \mathcal{O}\left(\sum_{i=1}^{+\infty} \mathbb{E}[\mathbb{I}^{(i)}]\epsilon_{\tau_{3i-1,T}}^{m-1}\right) + \mathcal{O}\left(\sum_{i=1}^{+\infty} \mathbb{E}\left[\mathbb{I}^{(i)} \sum_{t=\tau_{3i-1,T}}^{\tau_{3i,T}-1} \epsilon_t^{m+1}\|g_t\|^2\right]\right)$$
$$+ \sum_{i=1}^{+\infty} R_i. \tag{4}$$

Note that when the indicator function $\mathbb{I}^{(i)} = 1$, it implies that $\tau_{3i-1,T} < \tau_{3i,T}$, which further means that $\tau_{3i-1}$ is not truncated by $T$ (since otherwise $\tau_{3i-1,T} = \tau_{3i,T} = T$), and that $f(\theta_{\tau_{3i-1}}) - f^*$ must strictly exceed $b$ (otherwise $\tau_{3i-1} = \tau_{3i}$). This implies that the sum on the left-hand side actually equals the sum of the $m$-order squared gradient variations when $1 \leq t < T$ and $f(\theta_t) - f^* > b$. Thus, we can conclude (informal):

$$\sum_{t=1}^{T} \mathbb{E}\left[\mathbb{I}_{[f(\theta_t)-f^*>b]}\epsilon_t^m \|\nabla f(\theta_t)\|^2\right] \leq \mathcal{O}\left(\mathbb{E}\left[\mathbb{I}_{[a<f(\theta_t)-f^*\leq b]}\epsilon_t^{m+1}\|g_t\|^2\right]\right) + \mathcal{O}\left(\sum_{i=1}^{+\infty} \mathbb{E}[\mathbb{I}^{(i)}\epsilon_{\tau_{3i-1,T}}^{m-1}]\right)$$
$$+ \sum_{i=1}^{+\infty} R_i.$$

One of the main challenges in the proof lies in handling the term $\mathcal{O}\left(\sum_{i=1}^{+\infty} \mathbb{E}[\mathbb{I}^{(i)}\epsilon_{\tau_{3i-1,T}}^{m-1}]\right)$, which is difficult to summarize in just a few high-level remarks. Readers are referred to the full proof, presented from Eq. 24 to Eq. 28 in Appendix C.6, for a detailed explanation of how this term is treated. In short, through the use of some probabilistic techniques, we can ultimately bound this term as follows (informal):

$$\mathcal{O}\left(\sum_{i=1}^{+\infty} \mathbb{E}[\mathbb{I}^{(i)}\epsilon_{\tau_{3i-1,T}}^{m-1}]\right) \leq \mathcal{O}\left(\mathbb{E}\left[\mathbb{I}_{[a<f(\theta_t)-f^*\leq b]}\epsilon_t^{m-1}M_t^2\right]\right),$$

where $M_t := \epsilon_t \nabla f(\theta_t)^\top (\nabla f(\theta_t) - g_t)$. In the region $[a < f(\theta_t) - f^* < b]$, we can easily obtain that $\eta < \|\nabla f(\theta_t)\| \leq \sqrt{2LC_p}$. Thus, we can further apply the weak growth condition (Assumption 3.2∼(b)) to derive the following when $[a < f(\theta_t) - f^* < b]$:

$$\mathbb{E}\left[\|g\|^2 \mid \mathscr{F}_{t-1}\right] \leq G\left(1 + \frac{1}{\eta^2}\right)\|\nabla f(\theta_t)\|^2,$$

and

$$\mathbb{E}\left[M_t^2 \mid \mathscr{F}_{t-1}\right] \leq 2LC_pG\left(1 + \frac{1}{\eta^2}\right)\epsilon_t^2\|\nabla f(\theta_t)\|^2.$$

Based on the above steps, we can ultimately derive a recursive inequality between the $m$-order squared gradient variation and the $m + 1$-order squared gradient variation.

$$\sum_{t=1}^{T} \mathbb{E}\left[\mathbb{I}_{[f(\theta_t)-f^*>b]}\epsilon_t^m \|\nabla f(\theta_t)\|^2\right] \leq \mathcal{O}\left(\mathbb{E}\left[\mathbb{I}_{[a<f(\theta_t)-f^*\leq b]}\epsilon_t^{m+1}\|\nabla f(\theta_t)\|^2\right]\right) + \sum_{i=1}^{+\infty} R_i.$$

Using this recursive relation, we can ultimately iterate the squared gradient variation up to the $\lceil p \rceil$-order squared gradient variation. The $\lceil p \rceil$-order squared gradient variation can then be straightforwardly shown to be bounded due to the step-size condition $\sum_{t=1}^{+\infty} \epsilon_t^p < +\infty$. In this way, we complete the proof of the boundedness of the squared variation. At this point, we have effectively proven that $\mathbb{E}\left(\sup_{1\leq t<T}(f(\theta_t) - f^*)\right)$ is bounded by a constant independent of $T$ (the details of this independence can be found in Appendix C.8, where the derivation of the relevant constants is provided). Letting $T \to +\infty$ and applying the *Lebesgue's Monotone Convergence* theorem, we obtain that $\mathbb{E}\left(\sup_{t\geq 1} f(\theta_t) - f^*\right)$ is also controlled by this bound. With this, we have proven Lemma 4.2 and 4.1. We can now proceed to prove that the trajectory of SGD is an asymptotic pseudotrajectory of the corresponding gradient flow.

## 4.4 Proof that the Trajectory of SGD is an Asymptotic Pseudotrajectory of the Corresponding Gradient Flow

As before, we first present this result in the form of a lemma:

**Lemma 4.3.** *Let $\{\theta_t\}_{t\geq 1} \subset \mathbb{R}^d$ be the sequence generated by the Stochastic Gradient Descent (SGD) algorithm with initial point $\theta_0$. Under Assumption 3.1, excluding Item (c), and Assumption 3.2 (for $p > 3$, refer to Item (c); for $2 < p \leq 3$, refer to Item (d)), and with the step sizes sequence $\{\epsilon_t\}_{t\geq 1}$ satisfying Setting 1, the trajectory of SGD is an asymptotic pseudotrajectory of the corresponding gradient flow almost surely.*

With Lemma 4.2 and 4.1 already proven, the proof becomes straightforward. We now present the complete proof here.

*Proof.* According to Lemma 4.1, it is easy to see that $\sup_{t\geq 1}(f(\theta_t) - f^*) < +\infty$ a.s. From Lemma B.1, we can easily deduce that $\sup_{t\geq 1}\|\nabla f(\theta_t)\|^2 \leq 2L \sup_{t\geq 1}(f(\theta_t) - f^*) < +\infty$ a.s. Therefore, it is straightforward to verify that Condition A1 in Property 1 is satisfied. For condition A2, we proceed by partitioning the analysis. Specifically, we have:

$$
\lim_{t\to+\infty} \Delta(t,T) = \lim_{t\to+\infty} \sup_{k\in(t,m(\tau_t+T)]} \left\{ \left\| \sum_{i=t}^{k-1} \mathbb{I}_{(i)}\epsilon_i(\nabla f(\theta_i) - g_i) \right\| + \left\| \sum_{i=t}^{k-1} \mathbb{I}_{(i)}^c\epsilon_i(\nabla f(\theta_i) - g_i) \right\| \right\}
$$

$$
\leq \underbrace{\lim_{t\to+\infty} \sup_{k\in(t,m(\tau_t+T)]} \left\{ \left\| \sum_{i=t}^{k-1} \mathbb{I}_{(i)}\epsilon_i(\nabla f(\theta_i) - g_i) \right\| \right\}}_{\Delta'(t,T)} + \underbrace{\lim_{t\to+\infty} \sup_{k\in(t,m(\tau_t+T)]} \left\{ \left\| \sum_{i=t}^{k-1} \mathbb{I}_{(i)}^c\epsilon_i(\nabla f(\theta_i) - g_i) \right\| \right\}}_{\Delta''(t,T)}.
$$

where, when $p > 3$, $\mathbb{I}_{(i)}$ is defined as the indicator function of the event $[f(\theta_t) - f^* \leq C_p]$, and when $2 \leq p < 3$, $\mathbb{I}_{(i)}$ is defined as the indicator function of the event $[\|\nabla f(\theta_i)\| < x]$. Next, we handle $\Upsilon_1$ and $\Upsilon_2$ separately. For $\Upsilon_1$, we know that within the event where $\mathbb{I}_{(i)} = 1$, the higher-order moments are bounded. Thus, by applying *Burkholder's* inequality, we obtain:

$$
\mathbb{E}\left[(\Delta'_{t,T})^{2p-2}\right] \overset{\textit{Burkholder's} \text{ inequality}}{\leq} C_{2p-2}\, \mathbb{E}\left[ \left( \sum_{i=t}^{m(\tau_t+T)-1} \mathbb{I}_{(i)}\epsilon_i^2\|\nabla f(\theta_i) - g_i\|^2 \right)^{p-1} \right]
$$

$$
\overset{\textit{Hölder's} \text{ inequality}}{\leq} C_{2p-2}\, \mathbb{E}\left[ \left( \sum_{i=t}^{m(\tau_t+T)-1} \epsilon_i \right)^{p-2} \sum_{i=t}^{m(\tau_t+T)-1} \mathbb{I}_{(i)}\epsilon_i^p\|\nabla f(\theta_i) - g_i\|^{2p-2} \right]
$$

$$
\leq C_{2p-2}T M_p^{\frac{2p-2}{p}} \sum_{i=t}^{m(\tau_t+T)-1} \epsilon_i^p,
$$

where $C_{2p-2}$ is a constant depend on $2p - 2$. From the preceding inequality, we get that

$$
\sum_{k=1}^{+\infty} \mathbb{E}\left[(\Delta'_{kT,T})^{2p-2}\right] \leq C_{2p-2}T M_p^{\frac{2p-2}{p}} \sum_{i=1}^{+\infty} \epsilon_i^p < +\infty.
$$

By the *Borel-Cantelli* lemma this proves that $\lim_{k\to+\infty} \Delta'_{kT,T} = 0$, a.s., that is $\lim_{t\to+\infty} \Delta'_{t,T} = 0$ a.s. Next, we handle $\Delta''_{t,T}$. It is easy to see that when the event corresponding to $\mathbb{I}_{(i)}^c$ occurs, $\|\nabla f(\theta_i)\|$ must have a lower bound. When $p > 3$, this lower bound is $\eta$, and when $2 < p \leq 3$, the lower bound is $x$. We unify these cases by defining the lower bound as $l_p$. Then, we can proceed to compute:

$$
\mathbb{E}\left[(\Delta''_{t,T})^2\right] \overset{\textit{Doob's} \text{ inequality}}{\leq} 4\,\mathbb{E}\left[ \sum_{i=t}^{m(\tau_t+T)-1} \mathbb{I}_{(i)}\epsilon_i^2\|\nabla f(\theta_i) - g_i\|^2 \right]
$$

$$
\overset{\textit{weak growth} \text{ condition with lower bound } l_p}{\leq} 4G\left(1 + \frac{1}{l_p^2}\right) \mathbb{E}\left[ \sum_{i=t}^{m(\tau_t+T)-1} \mathbb{I}_{(i)}\epsilon_i^2\|\nabla f(\theta_i)\|^2 \right],
$$

that is

$$\sum_{k=1}^{+\infty} \mathbb{E}\left[\left(\Delta_{kT,T}''\right)^2\right] \leq 4G\left(1+\frac{1}{l_p^2}\right)\sum_{i=1}^{+\infty}\mathbb{E}\left[\mathbb{I}_{(i)}\epsilon_i^2\|\nabla f(\theta_i)\|^2\right]$$

$$\leq \begin{cases} 4G\epsilon_1\left(1+\frac{1}{l_p^2}\right)\sum_{i=1}^{+\infty}\mathbb{E}\left[\mathbb{I}_{(i)}\epsilon_i\|\nabla f(\theta_i)\|^2\right], & \text{if } p > 3 \\ 4G\left(1+\frac{1}{l_p^2}\right)\sum_{i=1}^{+\infty}\mathbb{E}\left[\epsilon_i^2\|\nabla f(\theta_i)\|^2\right], & \text{if } 2 < p \leq 3. \end{cases}$$

$$\overset{\text{Lemma 4.2 and Lemma B.4}}{<} +\infty$$

By the *Borel-Cantelli* lemma this proves that $\lim_{k\to+\infty}\Delta_{kT,T}'' = 0$, a.s., that is $\lim_{t\to+\infty}\Delta_{t,T}'' = 0$ a.s. In conclusion, we can prove that $\lim_{t\to+\infty}\Delta(t,T) = 0$ a.s., which verifies Condition A2 in Property 1. Thus, we have proven that the trajectory of SGD is an asymptotic pseudotrajectory of the corresponding gradient flow almost surely.

The combination of these results leads directly to the proof of Theorem 3.1

$\square$

### 4.5 PROOF OF THEOREM 3.1

*Proof.* Under the given assumptions, $f$ serves as a strict Lyapunov function for gradient flow, as defined in Benaïm (2006). Specifically, this implies that $f(\Phi_t(x))$ decreases monotonically in $t$ unless $x$ is a stationary point of gradient flow. Moreover, according to *Sard's* theorem (Sard, 1942; Bates, 1993), the set $f(\{\theta|\nabla f(\theta) = 0\})$ of critical values has Lebesgue measure zero and an empty topological interior. Consequently, by leveraging Theorem 5.7 and Proposition 6.4 from Benaïm Benaïm (2006), any precompact asymptotic pseudotrajectory of gradient flow converges to a connected component $\mathcal{X}^*$ where $f$ remains constant. Lemma 4.1 and corecivity in Assumption 3.1 ensures that the APTs of gradient flow induced by SGD are almost surely bounded, confirming our conclusion. $\square$

Finally, combined with the boundedness of the expected supremum as established in Lemma 4.1, we can immediately apply the *Lebesgue's Dominated Convergence* theorem to deduce $L_2$ convergence from almost sure convergence.

### 4.6 PROOF OF THEOREM 3.2

*Proof.* Based on Lemma 4.1, we can derive the following inequality:

$$\mathbb{E}\left[\sup_{t\geq 1}\|\nabla f(\theta_t)\|^2\right] \overset{\text{Lemma B.1}}{\leq} 2L\,\mathbb{E}\left[\sup_{t\geq 1}(f(\theta_t)-f^*)\right] \overset{\text{Lemma 4.1}}{<} +\infty.$$

Then, using the almost sure convergence from Theorem 3.1 and *Lebesgue's Dominated Convergence* theorem, we can establish the mean-square convergence result, i.e., $\lim_{n\to\infty}\mathbb{E}\|\nabla f(\theta_n)\|^2 = 0$. $\square$

## 5 OVERALL CONCLUSIONS

In this article, we employ a novel analytical method, called stopping time method, to explore the asymptotic convergence of the SGD algorithm under more relaxed step-size conditions, providing more step-size options with convergence guarantees for practical applications. This work is distinguished by its minimal set of required assumptions, thereby broadening the scope of SGD applications to practical scenarios where traditional assumptions may not apply. The underlying philosophy of the stopping time method could potentially serve as a template for proving the convergence of other related stochastic optimization algorithms, such as Adaptive Moment Estimation (ADAM) Kingma (2014) and Stochastic Gradient Descent with Momentum (SGDM) Gitman et al. (2019), etc.

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

## A STOCHASTIC APPROXIMATION

**Stochastic Approximation** (Benaïm (2006)): Let $F : \mathbb{R}^d \to \mathbb{R}^d$ be a continous map. Consider here a discrete time process $\{x_t\}_{t \geq 1}$ living in $\mathbb{R}^d$ (an algorithm) whose general for, can be written as

$$x_{t+1} - x_t = \gamma_{t+1} \left( F(x_t) + U_{t+1} \right), \tag{5}$$

where $\{\gamma_t\}_{t \geq 1}$ is a given sequence of nonegative numbers such that $\sum_{t=1}^{+\infty} \gamma_t = +\infty$  $\lim_{t \to +\infty} \gamma_t = 0$, and $U_t \in \mathbb{R}^d$ are (deterministic or random) perturbation.

**Interpolated Process** (Benaïm (2006)): Let $\{x_t\}_{t \geq 1}$ be a sequence in $\mathbb{R}^d$, and let $\{\gamma_t\}_{t \geq 1}$ be a sequence of positive step sizes. Define the time sequence $\{\tau_t\}_{t \geq 1}$ by:

$$\tau_t = \sum_{k=1}^{t-1} \gamma_k, \quad \text{with } \tau_1 = 0.$$

We define the interpolated process $X : \mathbb{R}_+ \to \mathbb{R}^d$ and the piecewise constant process $\overline{X} : \mathbb{R}_+ \to \mathbb{R}^d$ by:

$$X(\tau_t + s) = x_t + s \frac{x_{t+1} - x_t}{\gamma_{t+1}}, \quad \text{and} \quad \overline{X}(\tau_t + s) = x_t,$$

for all $n \in \mathbb{N}$ and $0 \leq s < \gamma_{t+1}$. That is, we call $X$ the *interpolated process* of the sequence $\{x_t\}_{t \geq 1}$. The "inverse" mapping of $n \mapsto \tau_t$ is defined by the function $m : \mathbb{R}_+ \to \mathbb{N}$:

$$m(t) = \sup \left\{ k \geq 1 \,|\, t \geq \tau_k \right\}.$$

**Asymptotic Pesudotrajectory** (Benaïm (2006)): Given the ODE: $\dot{x} = F(x)$. A semiflow $\Phi$ on a metric space $(M, d)$ is a continuous map

$$\Phi : \mathbb{R}_+ \times M \to M,$$

$$(t, x) \to \Phi(t, x) = \Phi_t(x),$$

where $\Phi_t(x)$ represents the position at time $t$ of the solution to the ODE starting from $x$. Unlike previous methods, which are still based on the ODE approach within stochastic approximation, our analysis is entirely grounded in a probabilistic method based on stopping times. This provides a new perspective for addressing the convergence of SGD.

Then a continuous function $Y : \mathbb{R}_+ \to M$ is an asymptotic pesudotrajectory for this ODE if

$$\lim_{t \to +\infty} \sup_{0 \leq h \leq T} d(X(t+h), \Phi(X(t))) = 0$$

for any $T > 0$.

It is easy to observe that SGD is a standard stochastic approximation algorithm. We can express SGD in the following form:

$$\theta_{t+1} = \theta_t + \epsilon_t \left( -\nabla f(\theta_t) + \nabla f(\theta_t) - g_t \right),$$

where the corresponding parameter mappings are given by:

$$x_t = \theta_t, \quad F(x_t) = -\nabla f(\theta_t), \quad \gamma_{t+1} = \epsilon_t, \quad U_{t+1} = \nabla f(\theta_t) - g_t.$$

## B SUPPORTING LEMMAS

**Lemma B.1.** *Suppose that $f(x)$ is differentiable and lower bounded $f^* = \inf_{x \in \mathbb{R}^d} f(x) > -\infty$ and $\nabla f(x)$ is Lipschitz continuous with parameter $\mathcal{L} > 0$, then $\forall \, x \in \mathbb{R}^d$, we have*

$$\left\| \nabla f(x) \right\|^2 \leq 2\mathcal{L}\big(f(x) - f^*\big).$$

**Lemma B.2** (Descent Lemma). *Let $\{\theta_t\}$ be the sequence generated by the SGD. Under Assumption 3.1.1, the following inequality holds for all $t \geq 1$:*

$$f(\theta_{t+1}) - f(\theta_t) \leq -\epsilon_t \|\nabla f(\theta_t)\|^2 + \frac{L\epsilon_t^2}{2} \|g_t\|^2 + M_t, \tag{6}$$

*where $M_t := \epsilon_t \nabla f(\theta_t)^\top (\nabla f(\theta_t) - g_t)$.*

**Lemma B.3.** *Let $\{\theta_t\}_{t\geq 1} \subset \mathbb{R}^d$ be the sequence generated by SGD with step sizes $\{\epsilon_t\}_{t\geq 1}$. Under Assumption 3.1(b) (Lipschitz Continuous Gradient), and the assumptions on the stochastic gradient (Assumption 3.2), the following inequality holds for all integers $n \geq 1$, all time indices $t \geq 0$, and for all $y > 0$:*

$$\mathbb{I}_{\|\nabla f(\theta_t)\|^2 > y} \cdot \epsilon_t^m \|\nabla f(\theta_t)\|^2 \leq \mathbb{I}_{\|\nabla f(\theta_t)\|^2 > y} \cdot \epsilon_t^{m-1} \Delta_{f_t} + \mathbb{I}_{\|\nabla f(\theta_t)\|^2 > y} \cdot \epsilon_t^{m-1} \left(M_{t,1} + M_{t,2}\right)$$

$$+ \frac{LG}{2}\left(1 + \frac{1}{y}\right)\mathbb{I}_{\|\nabla f(\theta_t)\|^2 > y} \cdot \epsilon_t^{m+1}\|\nabla f(\theta_t)\|^2, \qquad (7)$$

*where:*

- $\Delta_{f_t} := f(\theta_t) - f(\theta_{t+1})$ *denotes the decrease in the loss function at iteration $t$.*

- $\mathbb{I}_A$ *is the indicator function of the event $A$, which equals 1 if $A$ is true and 0 otherwise.*

- $M_{t,1}$ *and $M_{t,2}$ are defined in Eq. 6.*

**Lemma B.4.** *Let $\{\theta_t\}_{t\geq 1} \subset \mathbb{R}^d$ be the sequence generated by SGD with initial point $\theta_0$ and a step-size $\{\epsilon_t\}_{t\geq 1}$ satisfying Setting 1 for $2 < p \leq 3$. Under Assumption 3.1, excluding Item (c), and Assumption 3.2, specifically using Item (d) for $2 < p \leq 3$, the following inequality holds:*

$$\sum_{t=1}^{T} \epsilon_t^{p-1} \mathbb{E}\left[\|\nabla f(\theta_t)\|^2\right] \leq C(p) < +\infty, \qquad (8)$$

*where $C(p)$ is a constant depending on $p$ and the initial step size $\epsilon_1$.*

**Lemma B.5.** *Let $\{\theta_t\}_{t\geq 1} \subset \mathbb{R}^d$ be the sequence generated by the Stochastic Gradient Descent (SGD) algorithm with the initial point $\theta_0$. Under Assumption 3.1, excluding Item (c), and Assumption 3.2 (when $p > 3$, refer to Item (c); when $2 < p \leq 3$, refer to Item (d)), and assuming the step sizes sequence $\{\epsilon_t\}_{t\geq 1}$ satisfies Setting 1, the following inequality holds for any $\delta_0 > 0$ (where for $2 < p \leq 3$, since Assumptions 3.2~(d) do not define $C_p$, we set $C_p := C_\eta + 1$ in this case):*

$$\sum_{t=1}^{+\infty} \mathbb{E}\left[\overline{\Delta}_{t,\delta_0}\right] \leq C(p, \delta_0) < +\infty.$$

*where*

$$\overline{\Delta}_{t,\delta_0} := \left(\mathbb{I}_{f(\theta_t) - f^* < C_p} |f(\theta_{t+1}) - f(\theta_t)| - \delta_0\right)_+,$$

*and $C(p, \delta_0)$ is a constant depending on $p$, $\delta_0$.*

**Lemma B.6.** *Let $\{\theta_t\}_{t\geq 1} \subset \mathbb{R}^d$ be the sequence generated by the Stochastic Gradient Descent (SGD) algorithm with the initial point $\theta_0$. Under Assumption 3.1, excluding Item (c), and Assumption 3.2 (when $p > 3$, refer to Item (c); when $2 < p \leq 3$, refer to Item (d)), and assuming the step sizes sequence $\{\epsilon_t\}_{t\geq 1}$ satisfies Setting 1, the following inequality holds for any $C_\eta < a < b < C_p$ (where for $2 < p \leq 3$, since Assumptions 3.2~(d) do not define $C_p$, we set $C_p := C_\eta + 1$ in this case):*

$$\sum_{t=1}^{T} \mathbb{E}\left[\mathbb{I}_{[f(\theta_t) - f^* > b]} \epsilon_t^m \|\nabla f(\theta_t)\|^2\right] \leq C_1(a,b) \sum_{t=1}^{T} \mathbb{E}\left[\mathbb{I}_{[a < f(\theta_t) - f^* \leq b]} \epsilon_t^{m+1} \|\nabla f(\theta_t)\|^2\right]$$

$$+ C_2(a,b), \qquad (9)$$

*where*

$$C_1(a,b) := G\left(1 + \frac{1}{\eta^2}\right)\left(\frac{2C_1 L}{b-a} + \frac{L}{2}\right) + G\left(1 + \frac{1}{\eta^2}\right)\frac{8LC_1}{(b-a)^2},$$

$$C_2(a,b) := aC_1\epsilon_1^{\frac{m-1}{2}} + \left(\epsilon_1^{\frac{m-1}{2}} + 1\right)C(p,(b-a)/2).$$

# C  PROOFS OF LEMMAS

## C.1  THE PROOF OF LEMMA B.1

*Proof.* For $\forall x \in \mathbb{R}^N$, we define function

$$g(t) = f\left(x + t\frac{x' - x}{\|x' - x\|}\right),$$

where $x'$ is a constant point such that $x' - x$ is parallel to $\nabla f(x)$. By taking the derivative, we obtain

$$g'(t) = \nabla_{x+t\frac{x'-x}{\|x'-x\|}} f\left(x + t\frac{x'-x}{\|x'-x\|}\right)^T \frac{x'-x}{\|x'-x\|}. \tag{10}$$

Through the Lipschitz condition of $\nabla f(x)$, we get $\forall t_1, t_2$

$$|g'(t_1) - g'(t_2)| = \left|\left(\nabla_{x+t_1\frac{x'-x}{\|x'-x\|}} f\left(x + t_1\frac{x'-x}{\|x'-x\|}\right) - \nabla_{x+t_2\frac{x'-x}{\|x'-x\|}} f\left(x + t_2\frac{x'-x}{\|x'-x\|}\right)\right)^T \frac{x'-x}{\|x'-x\|}\right|$$

$$\leq \left\|\nabla_{x+t_1\frac{x'-x}{\|x'-x\|}} f\left(x + t_1\frac{x'-x}{\|x'-x\|}\right) - \nabla_{x+t_2\frac{x'-x}{\|x'-x\|}} f\left(x + t_2\frac{x'-x}{\|x'-x\|}\right)\right\| \left\|\frac{x'-x}{\|x'-x\|}\right\| \leq \mathcal{L}|t_1 - t_2|.$$

So $g'(t)$ satisfies the Lipschitz condition, and we have $\inf_{t\in\mathbb{R}} g(t) \geq \inf_{x\in\mathbb{R}^N} f(x) > -\infty$. Let $g^* = \inf x \in_\mathbb{R} g(x)$, then it holds that for $\forall t_0 \in \mathbb{R}$,

$$g(0) - g^* \geq g(0) - g(t_0). \tag{11}$$

By using the *Newton-Leibniz's* formula, we get that

$$g(0) - g(t_0) = \int_{t_0}^0 g'(\alpha)d\alpha = \int_{t_0}^0 \left(g'(\alpha) - g'(0)\right)d\alpha + \int_{t_0}^0 g'(0)d\alpha.$$

Through the Lipschitz condition of $g'$, we get that

$$g(0) - g(t_0) \geq \int_{t_0}^0 -\mathcal{L}|\alpha - 0|d\alpha + \int_{t_0}^0 g'(0)d\alpha = \frac{1}{2\mathcal{L}}\left(g'(0)\right)^2.$$

Then we take a special value of $t_0$. Let $t_0 = -g'(0)/\mathcal{L}$, then we get

$$g(0) - g(t_0) \geq -\int_{t_0}^0 \mathcal{L}|\alpha|d\alpha + \int_{t_0}^0 g(0)dt = -\frac{\mathcal{L}}{2}(0 - t_0)^2 + g'(0)(-t_0)$$

$$= -\frac{1}{2\mathcal{L}}\left(g'(0)\right)^2 + \frac{1}{\mathcal{L}}\left(g'(0)\right)^2 = \frac{1}{2\mathcal{L}}\left(g'(0)\right)^2. \tag{12}$$

Substituting Eq. 12 into Eq. 11, we get

$$g(0) - g^* \geq \frac{1}{2\mathcal{L}}\left(g'(0)\right)^2.$$

Due to $g^* \geq f^*$ and $\left(g'(0)\right)^2 = \|\nabla f(x)\|^2$, it follows that

$$\left\|\nabla f(x)\right\|^2 \leq 2\mathcal{L}\left(f(x) - f^*\right).$$

$\square$

## C.2 THE PROOF OF LEMMA B.2

*Proof.* We compute $f(\theta_{t+1}) - f(\theta_t)$. According to the $L$-smooth condition, we obtain the following estimate:

$$f(\theta_{t+1}) - f(\theta_t) \leq \nabla f(\theta_t)^\top (\theta_{t+1} - \theta_t) + \frac{L}{2}\|\theta_{t+1} - \theta_t\|^2$$

$$= -\epsilon_t \nabla f(\theta_t)^\top g_t + \frac{L\epsilon_t^2}{2}\|g_t\|^2$$

$$= -\epsilon_t \|\nabla f(\theta_t)\|^2 + \underbrace{\epsilon_t \nabla f(\theta_t)^\top (\nabla f(\theta_t) - g_t)}_{M_t} + \frac{L\epsilon_t^2}{2}\|g_t\|^2.$$

We complete the proof. $\square$

## C.3 THE PROOF OF LEMMA B.3

*Proof.* For any $y > 0$, we multiply both sides of Eq. 6 by the indicator function $\mathbb{I}_{\|\nabla f(\theta_t)\|^2 > y}$. This represents considering the derived properties of Eq. 6 when the event $\{\|\nabla f(\theta_t)\|^2 > y\}$ occurs. Specifically, we have:

$$\mathbb{I}_{\|\nabla f(\theta_t)\|^2 > y} \big( \underbrace{f(\theta_{t+1}) - f(\theta_t)}_{-\Delta_{f_t}} \big) \leq -\mathbb{I}_{\|\nabla f(\theta_t)\|^2 > y} \cdot \epsilon_t \|\nabla f(\theta_t)\|^2 + \mathbb{I}_{\|\nabla f(\theta_t)\|^2 > y} M_t$$

$$+ \mathbb{I}_{\|\nabla f(\theta_t)\|^2 > y} \frac{L\epsilon_t^2}{2} \|g_t\|^2. \tag{13}$$

After simplification, we obtain:

$$\mathbb{I}_{\|\nabla f(\theta_t)\|^2 > y} \cdot \epsilon_t \|\nabla f(\theta_t)\|^2 \leq \mathbb{I}_{\|\nabla f(\theta_t)\|^2 > y} \Delta_{f_t} + \mathbb{I}_{\|\nabla f(\theta_t)\|^2 > y} M_t$$

$$+ \mathbb{I}_{\|\nabla f(\theta_t)\|^2 > y} \frac{L\epsilon_t^2}{2} \|g_t\|^2. \tag{14}$$

For any $n \geq 1$, we multiply both sides of the above inequality by $\epsilon_t^{m-1}$, and we obtain:

$$\mathbb{I}_{\|\nabla f(\theta_t)\|^2 > y} \cdot \epsilon_t^m \|\nabla f(\theta_t)\|^2 \leq \mathbb{I}_{\|\nabla f(\theta_t)\|^2 > y} \epsilon_t^{m-1} \Delta_{f_t} + \mathbb{I}_{\|\nabla f(\theta_t)\|^2 > y} \epsilon_t^{m-1} M_t$$

$$+ \mathbb{I}_{\|\nabla f(\theta_t)\|^2 > y} \frac{L\epsilon_t^{m+1}}{2} \|g_t\|^2. \tag{15}$$

$\square$

## C.4 THE PROOF OF LEMMA B.4

*Proof.* For this goal, we multiply both sides of the descent inequality obtained in Lemma B.2 by $\epsilon_t^{p-2}$, and noting that $\epsilon_t \geq \epsilon_{t+1}$, we obtain:

$$\epsilon_{t+1}^{p-2}(f(\theta_{t+1}) - f^*) - \epsilon_t^{p-2}(f(\theta_t) - f^*) \leq -\epsilon_t^{p-1} \|\nabla f(\theta_t)\|^2 + \frac{L\epsilon_t^p}{2} \|g_t\|^2 + \epsilon_t^{p-2} M_t.$$

We take the expectation on both sides of the above inequality and then sum over the index $t$ from 1 to $+\infty$, which gives us:

$$\sum_{t=1}^{+\infty} \epsilon_t^{p-1} \mathbb{E}\left[\|\nabla f(\theta_t)\|^2\right] \leq \epsilon_1^{p-2}(f(\theta_1) - f^*) - \sum_{t=1}^{+\infty} \epsilon_t^{p-1} \mathbb{E}\|\nabla f(\theta_t)\|^2 + \frac{LG}{2}\sum_{t=1}^{+\infty} \epsilon_t^p (\mathbb{E}\|\nabla f(\theta_t)\|^2 + 1)$$

$$\leq \epsilon_1^{p-2}(f(\theta_1) - f^*) + \frac{LG}{2}\sum_{t=1}^{T_0} \epsilon_t^p \mathbb{E}\|\nabla f(\theta_t)\|^2 + \frac{LG}{2}\sum_{t=1}^{+\infty} \epsilon_t^p$$

$$< +\infty,$$

where $T_0$ is an index representing the largest index satisfying $\epsilon_t^{p-1} < \frac{LG}{2}\epsilon_t^p$. It is clear from $\lim_{t \to +\infty} \epsilon_t = 0$ that $T_0$ must exist and is a constant. With this, we prove this lemma. $\square$

## C.5 THE PROOF OF LEMMA B.5

*Proof.* We will prove this conclusion in two cases. First, we consider the case when $p > 3$. We have

$$\mathbb{I}_{f(\theta_t)-f^* < C_p} (f(\theta_{t+1}) - f(\theta_t)) = \mathbb{I}_{f(\theta_t)-f^* < C_p} \nabla f(\theta_{\xi_t})^\top (\theta_{t+1} - \theta_t)$$

$$= \mathbb{I}_{f(\theta_t)-f^* < C_p} \nabla f(\theta_t)^\top (\theta_{t+1} - \theta_t)$$

$$+ \mathbb{I}_{f(\theta_t)-f^* < C_p} (\nabla f(\theta_{\xi_t}) - \nabla f(\theta_t))^\top (\theta_{t+1} - \theta_t).$$

Taking absolute values on both sides, we obtain:

$$\mathbb{I}_{f(\theta_t)-f^* < C_p} |f(\theta_{t+1}) - f(\theta_t)| \leq \mathbb{I}_{f(\theta_t)-f^* < C_p} \|\nabla f(\theta_t)\| \cdot \|\theta_{t+1} - \theta_t\|$$

$$+ \mathbb{I}_{f(\theta_t)-f^* < C_p} \|\nabla f(\theta_{\xi_t}) - \nabla f(\theta_t)\| \cdot \|\theta_{t+1} - \theta_t\|$$

$$\overset{\text{L-Smooth}}{\leq} \mathbb{I}_{f(\theta_t)-f^*<C_p}\epsilon_t\|\nabla f(\theta_t)\|\|g_t\| + \mathbb{I}_{f(\theta_t)-f^*<C_p}L\epsilon_t^2\|g_t\|^2$$

$$\overset{\text{Lemma B.1}}{\leq} \mathbb{I}_{f(\theta_t)-f^*<C_p}\epsilon_t\sqrt{2LC_p}\|g_t\| + \mathbb{I}_{f(\theta_t)-f^*<C_p}L\epsilon_t^2\|g_t\|^2$$

$$\overset{\textit{AM-GM} \text{ inequality}}{\leq} \frac{\delta_0}{2} + \left(1+\frac{C_p}{\delta_0}\right)L\mathbb{I}_{f(\theta_t)-f^*<C_p}\epsilon_t^2\|g_t\|^2. \tag{16}$$

Next, we apply *Young's* inequality to continue the expansion, which gives us:

$$\mathbb{I}_{f(\theta_t)-f^*<C_p}|f(\theta_{t+1})-f(\theta_t)| \leq \frac{3}{4}\delta_0 + C_{\delta_0}\mathbb{I}_{f(\theta_t)-f^*<C_p}\epsilon_t^p\|g_t\|^p$$

$$< \delta_0 + C_{\delta_0}\mathbb{I}_{f(\theta_t)-f^*<C_p}\epsilon_t^p\|g_t\|^p,$$

where

$$C_{\delta_0} := \left(1+\frac{C_p}{\delta_0}\right)L \cdot \frac{2}{p}\left(\frac{4\left(1+\frac{C_p}{\delta_0}\right)L\left(1-\frac{2}{p}\right)}{\delta_0}\right)^{\frac{p-2}{2}},$$

that is

$$\left(\mathbb{I}_{f(\theta_t)-f^*<C_p}|f(\theta_{t+1})-f(\theta_t)| - \delta_0\right)_+ \leq C_{\delta_0}\mathbb{I}_{f(\theta_t)-f^*<C_p}\epsilon_t^p\|g_t\|^p. \tag{17}$$

Based on Setting 1, we can conclude that

$$\sum_{t=1}^{+\infty}\mathbb{E}\left[\left(\mathbb{I}_{f(\theta_t)-f^*<C_p}|f(\theta_{t+1})-f(\theta_t)| - \delta_0\right)_+\right] \leq C_{\delta_0}M_p\sum_{t=1}^{+\infty}\epsilon_t^p < +\infty.$$

Now, let us consider the case when $2 < p \leq 3$. In this scenario, the assumptions we use are Assumption 3.2 $\sim$ (c'). Under this assumption, through Lemma B.4, we have:

$$\sum_{t=1}^{T}\epsilon_t^{p-1}\mathbb{E}\left[\|\nabla f(\theta_t)\|^2\right] \leq C(p) := C(p,\delta_0) < +\infty, \tag{18}$$

where $C(p)$ is a constant depend on $p$. Then we use the results from Eq. 16 and continue the derivation, yielding:

$$\mathbb{I}_{f(\theta_t)-f^*<C_p}|f(\theta_{t+1})-f(\theta_t)| \leq \frac{\delta_0}{2} + \left(1+\frac{C_p}{\delta_0}\right)L\mathbb{I}_{f(\theta_t)-f^*<C_p}\epsilon_t^2\|g_t\|^2$$

$$= \frac{\delta_0}{2} + \left(1+\frac{C_p}{\delta_0}\right)L\mathbb{I}_{[f(\theta_t)-f^*<C_p]\cap[\|\nabla f(\theta_t)\|<\epsilon]}\epsilon_t^2\|g_t\|^2$$

$$+ \left(1+\frac{C_p}{\delta_0}\right)L\mathbb{I}_{[f(\theta_t)-f^*<C_p]\cap[\|\nabla f(\theta_t)\|\geq\epsilon]}\epsilon_t^2\|g_t\|^2$$

$$\leq \delta_0 + \left(1+\frac{C_p}{\delta_0}\right)L\mathbb{I}_{[f(\theta_t)-f^*<C_p]\cap[\|\nabla f(\theta_t)\|<\epsilon]}\epsilon_t^2\|g_t\|^2$$

$$+ C_{\delta_0}\mathbb{I}_{[f(\theta_t)-f^*<C_p]\cap[\|\nabla f(\theta_t)\|\geq\epsilon]}\epsilon_t^p\|g_t\|^p. \tag{19}$$

Then we can get

$$\sum_{t=1}^{+\infty}\mathbb{E}\left[\left(\mathbb{I}_{f(\theta_t)-f^*<C_p}|f(\theta_{t+1})-f(\theta_t)| - \delta_0\right)_+\right] \leq \left(1+\frac{C_p}{\delta_0}\right)\left(1+\frac{1}{x^2}\right)LC(p) + C_{\delta_0}M_p\sum_{t=1}^{+\infty}\epsilon_t^p$$

$$:= C(p,\delta_0) < +\infty.$$

With this, we complete the proof.

$\square$

## C.6    THE PROOF OF LEMMA B.6

*Proof.* For any $C_\eta < a < b < C_p$ (where we set $C_p = C_\eta + 1$ if $2 < p \leq 3$), we construct the following stopping time sequence based on the relationship between $f(\theta_t) - f^*$ and the positions of $a$ and $b$:

$$\tau_1 := \min\{t \geq 1 : f(\theta_t) - f^* > a\}, \ \tau_2 := \min\{t \geq \tau_1 : f(\theta_t) - f^* > b \text{ or } f(\theta_t) - f^* \leq a\},$$

$$\tau_3 := \min\{t \geq \tau_2 : f(\theta_t) - f^* \leq a\}, ...,$$

$$\tau_{3k-2} := \min\{t \geq \tau_{3k-3} : f(\theta_t) - f^* > a\},$$

$$\tau_{3k-1} := \min\{t \geq \tau_{3k-2} : f(\theta_t) - f^* > b \text{ or } f(\theta_t) - f^* \leq a\},$$

$$\tau_{3k} := \min\{t \geq \tau_{3k-1} : f(\theta_t) - f^* \leq a\}.$$

Next, for any $T \geq 1$, we define the truncated stopping time as $\tau_{n,T} := \tau_n \wedge T$. Then, by applying Lemma B.3 with $x = \eta$ on the interval $[\tau_{3i-1,T}, \tau_{3i,T})$ when $\tau_{3i-1,T} < \tau_{3i,T}$, we obtain that $\forall t \in [\tau_{3i-1,T}, \tau_{3i,T})$ when $\tau_{3i-1,T} < \tau_{3i,T}$, there is:

$$\mathbb{I}^{(i)}\mathbb{I}_{\|\nabla f(\theta_t)\|^2 > \eta} \cdot \epsilon_t^m \|\nabla f(\theta_t)\|^2 \leq \mathbb{I}^{(i)}\mathbb{I}_{\|\nabla f(\theta_t)\|^2 > \eta}\epsilon_t^{m-1}\Delta_{f_t} + \mathbb{I}^{(i)}\mathbb{I}_{\|\nabla f(\theta_t)\|^2 > \eta}\epsilon_t^{m-1}M_t$$

$$+ \mathbb{I}^{(i)}\mathbb{I}_{\|\nabla f(\theta_t)\|^2 > \eta}\frac{L\epsilon_t^{m+1}}{2}\|g_t\|^2,$$

where $\mathbb{I}^{(i)} := \mathbb{I}_{[\tau_{3i-1,T} < \tau_{3i,T}]}$. Summing the indices $t$ in the above inequality from $\tau_{3i-1,T}$ to $\tau_{3i,T}-1$ under the event $[\tau_{3i-1,T} < \tau_{3i,T}]$, we obtain:

$$\mathbb{I}^{(i)} \sum_{t=\tau_{3i-1,T}}^{\tau_{3i,T}-1} \mathbb{I}_{\|\nabla f(\theta_t)\|^2 > \eta} \cdot \epsilon_t^m \|\nabla f(\theta_t)\|^2 \leq \mathbb{I}^{(i)} \sum_{t=\tau_{3i-1,T}}^{\tau_{3i,T}-1} \mathbb{I}_{\|\nabla f(\theta_t)\|^2 > \eta}\epsilon_t^{m-1}\Delta_{f_t}$$

$$+ \mathbb{I}^{(i)} \sum_{t=\tau_{3i-1,T}}^{\tau_{3i,T}-1} \mathbb{I}_{\|\nabla f(\theta_t)\|^2 > \eta}\epsilon_t^{m-1}M_t$$

$$+ \frac{L}{2}\mathbb{I}^{(i)} \sum_{t=\tau_{3i-1,T}}^{\tau_{3i,T}-1} \mathbb{I}_{\|\nabla f(\theta_t)\|^2 > \eta} \cdot \epsilon_t^{m+1}\|g_t\|^2.$$

It is easy to see that the event $[\tau_{3i-1,T} < \tau_{3i,T}]$ is equivalent to $[\tau_{3i-1} < T] \cap [\tau_{3i-1} < \tau_{3i}]$. Therefore, when $\mathbb{I}_{[\tau_{3i-1,T} < \tau_{3i,T}]} = 1$, we have $\mathbb{I}_{\|\nabla f(\theta_t)\| > \eta} = 1$ for all $t \in [\tau_{3i-1,T}, \tau_{3i,T})$. As a result, we can remove the indicator function $\mathbb{I}_{\|\nabla f(\theta_t)\| > \eta}$ from both sides of the above inequality, yielding:

$$\mathbb{I}^{(i)} \sum_{t=\tau_{3i-1,T}}^{\tau_{3i,T}-1} \epsilon_t^m \|\nabla f(\theta_t)\|^2 \leq \mathbb{I}^{(i)} \sum_{t=\tau_{3i-1,T}}^{\tau_{3i,T}-1} \epsilon_t^{m-1}\Delta_{f_t} + \mathbb{I}^{(i)} \sum_{t=\tau_{3i-1,T}}^{\tau_{3i,T}-1} \epsilon_t^{m-1}M_t$$

$$+ \frac{LG}{2}\Big(1 + \frac{1}{\eta}\Big)\mathbb{I}^{(i)} \sum_{t=\tau_{3i-1,T}}^{\tau_{3i,T}-1} \epsilon_t^{m+1}\|\nabla f(\theta_t)\|^2$$

$$\overset{(a)}{\leq} \mathbb{I}^{(i)}\epsilon_{\tau_{3i-1,T}}^{m-1}(f(\theta_{\tau_{3i-1,T}}) - f^*) + \mathbb{I}^{(i)} \sum_{t=\tau_{3i-1,T}}^{\tau_{3i,T}-1} \epsilon_t^{m-1}M_t$$

$$+ \frac{L}{2}\mathbb{I}^{(i)} \sum_{t=\tau_{3i-1,T}}^{\tau_{3i,T}-1} \epsilon_t^{m+1}\|g_t\|^2. \tag{20}$$

In step $(a)$, the transformation is mainly applied to the first term on the left-hand side of the corresponding inequality. We have:

$$\mathbb{I}^{(i)} \sum_{t=\tau_{3i-1,T}}^{\tau_{3i,T}-1} \epsilon_t^{m-1}\Delta_{f_t} \leq \mathbb{I}^{(i)} \sum_{t=\tau_{3i-1,T}}^{\tau_{3i,T}-1} \left(\epsilon_t^{m-1}(f(\theta_t) - f^*) - \epsilon_{t+1}^{m-1}(f(\theta_{t+1}) - f^*)\right)$$

$$< \mathbb{I}^{(i)}\epsilon_{\tau_{3i-1,T}}^{m-1}(f(\theta_{\tau_{3i-1,T}}) - f^*).$$

Taking the expectation on both sides of Eq. 20, we obtain:

$$\mathbb{E}\left[\mathbb{I}^{(i)}\sum_{t=\tau_{3i-1,T}}^{\tau_{3i,T}-1}\epsilon_t^m\|\nabla f(\theta_t)\|^2\right]\leq\underbrace{\mathbb{E}\left[\mathbb{I}^{(i)}\epsilon_{\tau_{3i-1,T}-1}^{m-1}(f(\theta_{\tau_{3i-1,T}})-f^*)\right]}_{\Theta_{i,T,1}}$$

$$+\underbrace{\mathbb{E}\left[\mathbb{I}^{(i)}\sum_{t=\tau_{3i-1,T}}^{\tau_{3i,T}-1}\epsilon_t^{m-1}M_t\right]}_{\Theta_{i,T,2}}$$

$$+\frac{L}{2}\left[\mathbb{I}^{(i)}\sum_{t=\tau_{3i-1,T}}^{\tau_{3i,T}-1}\epsilon_t^{m+1}\|g_t\|^2\right].\tag{21}$$

First, let's handle $\Theta_{t,T,1}$. According to the definition of stopping times, when $\mathbb{I}_i=1$, that is, when $\tau_{3i-1,T}<\tau_{3i,T}$, we have $f(\theta_{\tau_{3i-1,T}-1})-f^*\leq b<C_p$, that means:

$$\Theta_{i,T,1}\leq b\,\mathbb{E}\left[\mathbb{I}^{(i)}\epsilon_{\tau_{3i-1,T}-1}^{m-1}\right]+\mathbb{E}\left[\overline{\Delta}_{\tau_{3i-1,T}-1,1}\right]$$

$$\leq(b+1)\,\mathbb{E}[\mathbb{I}^{(i)}]+\mathbb{E}\left[\overline{\Delta}_{\tau_{3i-1,T}-1,1}\right],\tag{22}$$

where $\overline{\Delta}_{\tau_{3i-1,T}-1,1}$ defined in Lemma B.5. Then we aim to address $\Theta_{i,T,2}$. Upon observation, it is evident that for any $n,k$, the stopping time $\tau_k$ satisfies the following additional property: $[\tau_k=n]\in\mathscr{F}_{n-1}$. This implies that the preceding time, $\tau_k-1$, is also a stopping time. Therefore, for $\Theta_{i,T,2}$.

$$\Theta_{i,T,2}=\mathbb{E}\left[\mathbb{E}\left[\mathbb{I}^{(i)}\sum_{t=\tau_{3i-1,T}}^{\tau_{3i,T}-1}\epsilon_t^{m-1}M_t\,\middle|\,\mathscr{F}_{\tau_{3i-1,T}-1}\right]\right]$$

$$=\mathbb{E}\left[\mathbb{I}^{(i)}\,\mathbb{E}\left[\sum_{t=\tau_{3i-1,T}}^{\tau_{3i,T}-1}\epsilon_t^{m-1}M_t\,\middle|\,\mathscr{F}_{\tau_{3i-1,T}-1}\right]\right]$$

$$\overset{\text{\textit{Doob's Stopped} theorem}}{=}\mathbb{E}\left[\mathbb{I}^{(i)}\,\mathbb{E}\left[\sum_{t=\tau_{3i-1,T}}^{\tau_{3i,T}-1}\epsilon_t^{m-1}\,\mathbb{E}[M_t|\mathscr{F}_{t-1}]\,\middle|\,\mathscr{F}_{\tau_{3i-1,T}-1}\right]\right]$$

$$=0$$

We combine the estimates related to $\Theta_{i,T,2}$ from the above expression with those related to $\Theta_{i,T,1}$ in Eq. 22, and substitute both back into Eq. 21, we obtain:

$$\mathbb{E}\left[\mathbb{I}^{(i)}\sum_{t=\tau_{3i-1,T}}^{\tau_{3i,T}-1}\epsilon_t^m\|\nabla f(\theta_t)\|^2\right]\leq C_1\,\mathbb{E}[\mathbb{I}^{(i)}\epsilon_{\tau_{3i-1},T}^{m-1}]+\mathbb{E}\left[\overline{\Delta}_{\tau_{3i-1,T}-1,1}\right]$$

$$+\frac{L}{2}\left[\mathbb{I}^{(i)}\sum_{t=\tau_{3i-1,T}}^{\tau_{3i,T}-1}\epsilon_t^{m+1}\|g_t\|^2\right],\tag{23}$$

where

$$C_1:=b+1.$$

Next, we will address the first term in the above inequality. Here, we examine the properties of $\mathbb{I}_i$. When $\mathbb{I}_i=1$, we have the event $[\tau_{3i-1,T}<\tau_{3i,T}]$. This implies that both $[\tau_{3i-1}<\tau_{3i}]$ and $[\tau_{3i-1}<T]$ hold simultaneously. Consequently, based on the definition of the stopping sequence, we obtain:

$$\mathbb{I}_i\big(f(\theta_{\tau_{3i-1,T}})-f^*\big)>\mathbb{I}_ib.$$

This implies that

$$\mathbb{I}_i\big(f(\theta_{\tau_{3i-1,T}})-f(\theta_{\tau_{3i-2,T}-1})\big)>\mathbb{I}_i(b-a).\tag{24}$$

That is to say,

$$\mathbb{I}_i\big(\epsilon_{\tau_{3i-1,T}}^{\frac{m-1}{2}}(f(\theta_{\tau_{3i-1,T}})-f^*)-\epsilon_{\tau_{3i-2,T}-1}^{\frac{m-1}{2}}(f(\theta_{\tau_{3i-2,T}-1})-f^*)\big) > \mathbb{I}_i\epsilon_{\tau_{3i-1,T}}^{m-1}(b-a).$$

Since $f(\theta_{\tau_{3i-2,T}-1})-f^* \le a$, we easily obtain:

$$\mathbb{I}_i\big(\epsilon_{\tau_{3i-1,T}}^{\frac{m-1}{2}}(f(\theta_{\tau_{3i-1,T}})-f^*)-\epsilon_{\tau_{3i-2,T}-1}^{\frac{m-1}{2}}(f(\theta_{\tau_{3i-2,T}-1})-f^*)+a(\epsilon_{\tau_{3i-2,T}-1}^{\frac{m-1}{2}}-\epsilon_{\tau_{3i-1,T}}^{\frac{m-1}{2}}))$$

$$\ge \mathbb{I}_i\big(\epsilon_{\tau_{3i-1,T}}^{\frac{m-1}{2}}(f(\theta_{\tau_{3i-1,T}})-f^*)-\epsilon_{\tau_{3i-1,T}}^{\frac{m-1}{2}}(f(\theta_{\tau_{3i-2,T}-1})-f^*))$$

$$> \mathbb{I}_i\epsilon_{\tau_{3i-1,T}}^{\frac{m-1}{2}}(b-a).$$

We use the *Descent Lemma* (Lemma B.2) to bound $\epsilon_{\tau_{3i-1,T}}^{\frac{m-1}{2}}(f(\theta_{\tau_{3i-1,T}})-f^*) - \epsilon_{\tau_{3i-2,T}-1}^{\frac{m-1}{2}}(f(\theta_{\tau_{3i-2,T}-1})-f^*)$, yielding:

$$\mathbb{I}_i\epsilon_{\tau_{3i-1,T}}^{\frac{m-1}{2}}(b-a) < \mathbb{I}_i\big(\epsilon_{\tau_{3i-1,T}}^{\frac{m-1}{2}}(f(\theta_{\tau_{3i-1,T}})-f^*)-\epsilon_{\tau_{3i-2,T}-1}^{\frac{m-1}{2}}(f(\theta_{\tau_{3i-2,T}-1})-f^*)$$

$$+ a(\epsilon_{\tau_{3i-2,T}-1}^{\frac{m-1}{2}}-\epsilon_{\tau_{3i-1,T}}^{\frac{m-1}{2}}))$$

$$< \mathbb{I}_i\sum_{t=\tau_{3i-2,T}}^{\tau_{3i-1,T}-1}\epsilon_t^{\frac{m-1}{2}}\big(f(\theta_{t+1})-f(\theta_t)\big)+a(\epsilon_{\tau_{3i-2,T}-1}^{\frac{m-1}{2}}-\epsilon_{\tau_{3i-1,T}}^{\frac{m-1}{2}})$$

$$+ \mathbb{I}_i\epsilon_{\tau_{3i-2,T}-1}^{\frac{m-1}{2}}\overline{\Delta}_{\tau_{3i-2,T}-1,(b-a)/2}+\mathbb{I}_i\epsilon_{\tau_{3i-1,T}}^{\frac{m-1}{2}}\frac{b-a}{2}.$$

We multiply both sides of the above inequality by $\epsilon_{\tau_{3i-1,T}}^{\frac{m-1}{2}}$, and noting $\epsilon_{\tau_{3i-1,T}}^{\frac{m-1}{2}} < \epsilon_1^{\frac{m-1}{2}}$, we get:

$$\frac{\mathbb{I}_i\epsilon_{\tau_{3i-1,T}}^{m-1}(b-a)}{2} < \mathbb{I}_i\epsilon_{\tau_{3i-1,T}}^{\frac{m-1}{2}}\sum_{t=\tau_{3i-2,T}}^{\tau_{3i-1,T}-1}\epsilon_t^{\frac{m-1}{2}}\big(f(\theta_{t+1})-f(\theta_t)\big)+\mathbb{I}_i\epsilon_1^{\frac{m-1}{2}}\overline{\Delta}_{\tau_{3i-2,T}-1,(b-a)/2}$$

$$+ \epsilon_1^{\frac{m-1}{2}}\mathbb{I}_i a(\epsilon_{\tau_{3i-2,T}-1}^{\frac{m-1}{2}}-\epsilon_{\tau_{3i-1,T}}^{\frac{m-1}{2}})$$

$$\overset{\text{Lemma B.2}}{<} \mathbb{I}_i\frac{L}{2}\sum_{t=\tau_{3i-2,T}}^{\tau_{3i-1,T}-1}\epsilon_t^{m+1}\|g_t\|^2+\mathbb{I}_i\epsilon_{\tau_{3i-1,T}}^{\frac{m-1}{2}}\sum_{t=\tau_{3i-2,T}-1}^{\tau_{3i-1,T}-1}\epsilon_t^{\frac{m-1}{2}}M_t$$

$$+ \epsilon_1^{\frac{m-1}{2}}\mathbb{I}_i a(\epsilon_{\tau_{3i-2,T}-1}^{\frac{m-1}{2}}-\epsilon_{\tau_{3i-1,T}}^{\frac{m-1}{2}})+\mathbb{I}_i\epsilon_1^{\frac{m-1}{2}}\overline{\Delta}_{\tau_{3i-2,T}-1,(b-a)/2}$$

$$\overset{\textit{AM-GM inequality}}{\le} \mathbb{I}_i\frac{L}{2}\sum_{t=\tau_{3i-2,T}}^{\tau_{3i-1,T}-1}\epsilon_t^{m+1}\|g_t\|^2+\mathbb{I}_i\frac{(b-a)\epsilon_{\tau_{3i-1,T}}^{m-1}}{4}$$

$$+ \mathbb{I}_i\frac{1}{b-a}\left(\sum_{t=\tau_{3i-2,T}}^{\tau_{3i-1,T}-1}\epsilon_t^{\frac{m-1}{2}}M_t\right)^2$$

$$+ \mathbb{I}_i a(\epsilon_{\tau_{3i-2,T}-1}^{\frac{m-1}{2}}-\epsilon_{\tau_{3i-1,T}}^{\frac{m-1}{2}})+\mathbb{I}_i\epsilon_1^{\frac{m-1}{2}}\overline{\Delta}_{\tau_{3i-2,T}-1,(b-a)/2}. \qquad (25)$$

This implies that the following equation holds:

$$\frac{\mathbb{I}_i\epsilon_{\tau_{3i-1,T}}^{m-1}(b-a)}{4} < \mathbb{I}_i\frac{L}{2}\sum_{t=\tau_{3i-2,T}}^{\tau_{3i-1,T}-1}\epsilon_t^{m+1}\|g_t\|^2+\mathbb{I}_i\frac{1}{b-a}\left(\sum_{t=\tau_{3i-2,T}}^{\tau_{3i-1,T}-1}\epsilon_t^{\frac{m-1}{2}}M_t\right)^2$$

$$+ \mathbb{I}_i a(\epsilon_{\tau_{3i-2,T}-1}^{\frac{m-1}{2}}-\epsilon_{\tau_{3i-1,T}}^{\frac{m-1}{2}})+\mathbb{I}_i\epsilon_1^{\frac{m-1}{2}}\overline{\Delta}_{\tau_{3i-2,T}-1,(b-a)/2}.$$

Taking the expectation on both sides of the above inequality and scaling the indicator function $\mathbb{I}_i$ on the right side to 1, we obtain:

$$\frac{b-a}{4}\,\mathbb{E}\left[\mathbb{I}_i\epsilon_{\tau_{3i-1,T}}^{m-1}\right] < \frac{L}{2}\,\mathbb{E}\left[\sum_{t=\tau_{3i-2,T}}^{\tau_{3i-1,T}-1}\epsilon_t^{m+1}\|g_t\|^2\right]+\frac{1}{b-a}\,\mathbb{E}\left[\sum_{t=\tau_{3i-2,T}-1}^{\tau_{3i-1,T}-1}\epsilon_t^{\frac{m-1}{2}}M_t\right]^2$$

$$+ a\, \mathbb{E}[\epsilon_{\tau_{3i-2,T}-1}^{\frac{m-1}{2}} - \epsilon_{\tau_{3i-1,T}}^{\frac{m-1}{2}}] + \epsilon_1^{\frac{m-1}{2}}\, \mathbb{E}\left[\overline{\Delta}_{\tau_{3i-2,T}-1,(b-a)/2}\right]$$

$$\leq \frac{L}{2}\, \mathbb{E}\left[\sum_{t=\tau_{3i-2,T}}^{\tau_{3i-1,T}-1} \epsilon_t^{m+1} \|g_t\|^2\right]$$

$$+ \frac{1}{b-a}\, \mathbb{E}\left[\sum_{t=1}^{\tau_{3i-1,T}-1} \epsilon_t^{\frac{m-1}{2}} M_t - \sum_{t=1}^{\tau_{3i-2,T}-1} \epsilon_t^{\frac{m-1}{2}} M_t\right]^2$$

$$+ a\, \mathbb{E}[\epsilon_{\tau_{3i-2,T}}^{\frac{m-1}{2}} - \epsilon_{\tau_{3i-1,T}}^{\frac{m-1}{2}}] + \epsilon_1^{\frac{m-1}{2}}\, \mathbb{E}\left[\overline{\Delta}_{\tau_{3i-2,T}-1,(b-a)/2}\right]$$

$$\overset{(a)}{\leq} \frac{L}{2}\, \mathbb{E}\left[\sum_{t=\tau_{3i-2,T}}^{\tau_{3i-1,T}-1} \epsilon_t^{m+1} \|g_t\|^2\right]$$

$$+ \frac{1}{b-a}\, \mathbb{E}\left[\sum_{t=1}^{\tau_{3i-1,T}-1} \epsilon_t^{m-1} M_t^2 - \sum_{t=1}^{\tau_{3i-2,T}-1} \epsilon_t^{m-1} M_t^2\right]$$

$$+ a\, \mathbb{E}[\epsilon_{\tau_{3i-2,T}-1}^{\frac{m-1}{2}} - \epsilon_{\tau_{3i-1,T}}^{\frac{m-1}{2}}] + \epsilon_1^{\frac{m-1}{2}}\, \mathbb{E}\left[\overline{\Delta}_{\tau_{3i-2,T}-1,(b-a)/2}\right]$$

$$= \frac{L}{2}\, \mathbb{E}\left[\sum_{t=\tau_{3i-2,T}}^{\tau_{3i-1,T}-1} \epsilon_t^{m+1} \|g_t\|^2\right] + \frac{1}{b-a}\, \mathbb{E}\left[\sum_{t=\tau_{3i-2,T}}^{\tau_{3i-1,T}-1} \epsilon_t^{m-1} M_t^2\right]$$

$$+ a\, \mathbb{E}[\epsilon_{\tau_{3i-2,T}-1}^{\frac{m-1}{2}} - \epsilon_{\tau_{3i-1,T}}^{\frac{m-1}{2}}] + \epsilon_1^{\frac{m-1}{2}}\, \mathbb{E}\left[\overline{\Delta}_{\tau_{3i-2,T}-1,(b-a)/2}\right]. \qquad (26)$$

Now let us explain step $(a)$. Since it has been shown earlier that $\tau_{n,T}-1$ is also a stopping time, we know from Doob's stopping theorem that the following stopped process

$$\left(\sum_{t=1}^{\tau_{n,T}-1} \epsilon_t^{\frac{m-1}{2}} M_t,\, \mathscr{F}_{\tau_{n,T}-1}\right),$$

is still a martingale. Thus, we can easily derive:

$$\frac{1}{b-a}\, \mathbb{E}\left[\sum_{t=1}^{\tau_{3i-1,T}-1} \epsilon_t^{\frac{m-1}{2}} M_t - \sum_{t=1}^{\tau_{3i-2,T}-1} \epsilon_t^{\frac{m-1}{2}} M_t\right]^2$$

$$= \frac{1}{b-a}\, \mathbb{E}\left[\sum_{t=1}^{\tau_{3i-1,T}-1} \epsilon_t^{\frac{m-1}{2}} M_t\right]^2 - \frac{1}{b-a}\, \mathbb{E}\left[\sum_{t=1}^{\tau_{3i-2,T}-1} \epsilon_t^{\frac{m-1}{2}} M_t\right]^2$$

$$= \frac{1}{b-a}\, \mathbb{E}\left[\sum_{t=1}^{\tau_{3i-1,T}-1} \epsilon_t^{m-1} M_t^2\right] - \frac{1}{b-a}\, \mathbb{E}\left[\sum_{t=1}^{\tau_{3i-2,T}-1} \epsilon_t^{m-1} M_t^2\right].$$

Substituting Eq 26 back into Eq. 23, we obtain:

$$\mathbb{E}\left[\mathbb{I}^{(i)} \sum_{t=\tau_{3i-1,T}}^{\tau_{3i,T}-1} \epsilon_t^m \|\nabla f(\theta_t)\|^2\right] \leq \frac{2C_1 L}{b-a}\, \mathbb{E}\left[\sum_{t=\tau_{3i-2,T}}^{\tau_{3i-1,T}-1} \epsilon_t^{m+1} \|g_t\|^2\right] + \frac{4C_1}{(b-a)^2}\, \mathbb{E}\left[\sum_{t=\tau_{3i-2,T}}^{\tau_{3i-1,T}-1} \epsilon_t^{m-1} M_t^2\right]$$

$$+ aC_1\, \mathbb{E}[\epsilon_{\tau_{3i-2,T}-1}^{\frac{m-1}{2}} - \epsilon_{\tau_{3i-1,T}}^{\frac{m-1}{2}}] + \epsilon_1^{\frac{m-1}{2}}\, \mathbb{E}\left[\overline{\Delta}_{\tau_{3i-2,T}-1,(b-a)/2}\right]$$

$$+ \mathbb{E}\left[\overline{\Delta}_{\tau_{3i-1,T}-1,1}\right] + \frac{L}{2}\left[\mathbb{I}^{(i)} \sum_{t=\tau_{3i-1,T}}^{\tau_{3i,T}-1} \epsilon_t^{m+1} \|g_t\|^2\right].$$

We sum both sides of the above inequality over the index $i$, yielding:

$$\sum_{i=1}^{+\infty} \mathbb{E}\left[\mathbb{I}^{(i)} \sum_{t=\tau_{3i-1,T}}^{\tau_{3i,T}-1} \epsilon_t^m \|\nabla f(\theta_t)\|^2\right] \leq \left(\frac{2C_1 L}{b-a} + \frac{L}{2}\right) \sum_{i=1}^{+\infty} \mathbb{E}\left[\sum_{t=\tau_{3i-2,T}}^{\tau_{3i-1,T}-1} \epsilon_t^{m+1} \|g_t\|^2\right]$$

$$+ \frac{4C_1}{(b-a)^2} \sum_{i=1}^{+\infty} \mathbb{E}\left[\sum_{t=\tau_{3i-2,T}}^{\tau_{3i-1,T}-1} \epsilon_t^{m-1} M_t^2\right]$$

$$+ aC_1\epsilon_1^{\frac{m-1}{2}} + \left(\epsilon_1^{\frac{m-1}{2}} + 1\right) C(p,(b-a)/2), \qquad (27)$$

where $C(p,(b-a)/2)$ shown in Lemma B.5. Noting the following identity:

$$\sum_{i=1}^{+\infty} \sum_{t=\tau_{3i-2,T}}^{\tau_{3i-1,T}-1} F_t = \sum_{t=1}^{T} \mathbb{I}_{[a<f(\theta_t)-f^*\leq b]} F_t,$$

we can simplify Eq. 27 to:

$$\sum_{t=1}^{T} \mathbb{E}\left[\mathbb{I}_{[f(\theta_t)-f^*>b]}\epsilon_t^m\|\nabla f(\theta_t)\|^2\right] \leq \left(\frac{2C_1 L}{b-a} + \frac{L}{2}\right) \sum_{t=1}^{T} \mathbb{E}\left[\mathbb{I}_{[a<f(\theta_t)-f^*\leq b]}\epsilon_t^{m+1}\|g_t\|^2\right]$$

$$+ \frac{4C_1}{(b-a)^2} \sum_{t=1}^{T} \mathbb{E}\left[\mathbb{I}_{[a<f(\theta_t)-f^*\leq b]}\epsilon_t^{m-1} M_t^2\right]$$

$$+ aC_1\epsilon_1^{\frac{m-1}{2}} + \left(\epsilon_1^{\frac{m-1}{2}} + 1\right) C(p,(b-a)/2). \qquad (28)$$

Since both $a$ and $b$ are greater than $C_\eta$, according to Assumption 3.1 $\sim$ (d), we know that when $a < f(\theta_t) - f^* \leq b$, it holds that $\|\nabla f(\theta_t)\|^2 > \eta^2$. Thus, we can transform the first two terms on the right side of the above inequality as follows:

$$\left(\frac{2C_1 L}{b-a} + \frac{L}{2}\right) \sum_{t=1}^{T} \mathbb{E}\left[\mathbb{I}_{[a<f(\theta_t)-f^*\leq b]}\epsilon_t^{m+1}\|g_t\|^2\right]$$

$$= G\left(1 + \frac{1}{\eta^2}\right)\left(\frac{2C_1 L}{b-a} + \frac{L}{2}\right) \sum_{t=1}^{T} \mathbb{E}\left[\mathbb{I}_{[a<f(\theta_t)-f^*\leq b]}\epsilon_t^{m+1}\|\nabla f(\theta_t)\|^2\right],$$

and

$$\frac{4C_1}{(b-a)^2} \sum_{t=1}^{T} \mathbb{E}\left[\mathbb{I}_{[a<f(\theta_t)-f^*\leq b]}\epsilon_t^{m-1} M_t^2\right]$$

$$\leq G\left(1 + \frac{1}{\eta^2}\right) \frac{8LC_1}{(b-a)^2} \sum_{t=1}^{T} \mathbb{E}\left[\mathbb{I}_{[a<f(\theta_t)-f^*\leq b]}\epsilon_t^{m+1}\|\nabla f(\theta_t)\|^2\right].$$

Substituting the above two inequalities back into Eq. 28, we obtain:

$$\sum_{t=1}^{T} \mathbb{E}\left[\mathbb{I}_{[f(\theta_t)-f^*>b]}\epsilon_t^m\|\nabla f(\theta_t)\|^2\right] \leq C_1(a,b) \sum_{t=1}^{T} \mathbb{E}\left[\mathbb{I}_{[a<f(\theta_t)-f^*\leq b]}\epsilon_t^{m+1}\|\nabla f(\theta_t)\|^2\right]$$

$$+ C_2(a,b), \qquad (29)$$

where

$$C_1(a,b) := G\left(1 + \frac{1}{\eta^2}\right)\left(\frac{2C_1 L}{b-a} + \frac{L}{2}\right) + G\left(1 + \frac{1}{\eta^2}\right)\frac{8LC_1}{(b-a)^2},$$

$$C_2(a,b) := aC_1\epsilon_1^{\frac{m-1}{2}} + \left(\epsilon_1^{\frac{m-1}{2}} + 1\right) C(p,(b-a)/2).$$

With this, we complete the proof.

$\square$

## C.7 The Proof of Lemma 4.2

*Proof.* Since $C_\eta < C_p$ (and when $2 < p \leq 3$, we set $C_p = C_\eta + 1$), we can always insert $\lceil p \rceil$ ($\lceil p \rceil = \min\{n \in \mathbb{Z} \mid n \geq p\}$) distinct and equidistant real numbers between $C_\eta$ and $C_p$ such

that $C_\eta < a_{\lceil p \rceil} < a_{\lceil p \rceil - 1} < \cdots < a_1 < C_p$. Now, using Lemma B.6 with the parameters set as $a = a_{k+1}$, $b = a_k$, and $m = k$, we obtain the following inequality:

$$\sum_{t=1}^{T} \mathbb{E}\left[ \mathbb{I}_{[f(\theta_t) - f^* > a_k]} \epsilon_t^k \|\nabla f(\theta_t)\|^2 \right] \leq C_1(a_{k+1}, a_k) \sum_{t=1}^{T} \mathbb{E}\left[ \mathbb{I}_{[a_{k+1} < f(\theta_t) - f^* \leq a_k]} \epsilon_t^{k+1} \|\nabla f(\theta_t)\|^2 \right]$$
$$+ C_2(a_{k+1}, a_k), \tag{30}$$

Since we clearly have

$$\sum_{t=1}^{T} \mathbb{E}\left[ \mathbb{I}_{[a_{k+1} < f(\theta_t) - f^* \leq a_k]} \epsilon_t^{k+1} \|\nabla f(\theta_t)\|^2 \right] \leq \sum_{t=1}^{T} \mathbb{E}\left[ \mathbb{I}_{[f(\theta_t) - f^* > a_{k+1}]} \epsilon_t^{k+1} \|\nabla f(\theta_t)\|^2 \right],$$

we can derive the following inequality:

$$\sum_{t=1}^{T} \mathbb{E}\left[ \mathbb{I}_{[f(\theta_t) - f^* > a_k]} \epsilon_t^k \|\nabla f(\theta_t)\|^2 \right] \leq C_1(a_{k+1}, a_k) \sum_{t=1}^{T} \mathbb{E}\left[ \mathbb{I}_{[f(\theta_t) - f^* > a_{k+1}]} \epsilon_t^{k+1} \|\nabla f(\theta_t)\|^2 \right]$$
$$+ C_2(a_{k+1}, a_k). \tag{31}$$

Next, we define

$$Q_k := \sum_{t=1}^{T} \mathbb{E}\left[ \mathbb{I}_{[f(\theta_t) - f^* > a_k]} \epsilon_t \|\nabla f(\theta_t)\|^2 \right].$$

It can be seen that Eq. 31 actually implies the following recursive inequality:

$$Q_k \leq C_1(a_{k+1}, a_k) Q_{k+1} + C_2(a_{k+1}, a_k). \tag{32}$$

By iterating the above recursive inequality from index $k = 1$ to $k = \lceil p \rceil - 2$, we obtain:

$$Q_1 \leq \left( \prod_{i=1}^{\lceil p \rceil - 2} C_1(a_{i+1}, a_i) \right) Q_{\lceil p \rceil - 1} + \sum_{k=1}^{\lceil p \rceil - 2} \left( \prod_{i=1}^{k} C_1(a_{i+1}, a_i) \right) C_2(a_{k+1}, a_k). \tag{33}$$

Finally, by setting $k = \lceil p \rceil - 1$ in Eq. 30, we obtain:

$$Q_{\lceil p \rceil - 1} \leq C_1(a_{\lceil p \rceil}, a_{\lceil p \rceil - 1}) \sum_{t=1}^{T} \mathbb{E}\left[ \mathbb{I}_{[a_{\lceil p \rceil} < f(\theta_t) - f^* \leq a_{\lceil p \rceil - 1}]} \epsilon_t^{\lceil p \rceil} \|\nabla f(\theta_t)\|^2 \right]$$
$$+ C_2(a_{\lceil p \rceil}, a_{\lceil p \rceil - 1})$$
$$\overset{\text{Lemma B.1}}{\leq} 2LC_p C_1(a_{\lceil p \rceil}, a_{\lceil p \rceil - 1}) \sum_{t=1}^{+\infty} \epsilon_t^{\lceil p \rceil} + C_2(a_{\lceil p \rceil}, a_{\lceil p \rceil - 1}). \tag{34}$$

Substituting the above result back into Eq. 33, we obtain:

$$Q_1 \leq \left( \prod_{i=1}^{\lceil p \rceil - 2} C_1(a_{i+1}, a_i) \right) \left( 2LC_p C_1(a_{\lceil p \rceil}, a_{\lceil p \rceil - 1}) \sum_{t=1}^{+\infty} \epsilon_t^{\lceil p \rceil} + C_2(a_{\lceil p \rceil}, a_{\lceil p \rceil - 1}) \right)$$
$$+ \sum_{k=1}^{\lceil p \rceil - 2} \left( \prod_{i=1}^{k} C_1(a_{i+1}, a_i) \right) C_2(a_{k+1}, a_k) := C(p, C_\eta, C_p) < +\infty. \tag{35}$$

Since $\sum_{t=1}^{+\infty} \epsilon_t^p < +\infty$, we conclude that $\sum_{t=1}^{+\infty} \epsilon_t^{\lceil p \rceil} < +\infty$. It can be seen that the right-hand side of Eq. 35 is a constant that depends only on $p$, $C_\eta$, and $C_p$. We denote this constant by $C(p, C_\eta, C_p)$. Next, we revert to the original expression for $Q_1$, and we obtain:

$$\sum_{t=1}^{T} \mathbb{E}\left[ \mathbb{I}_{[f(\theta_t) - f^* > a_1]} \epsilon_t \|\nabla f(\theta_t)\|^2 \right] < C(p, C_\eta, C_p).$$

Next, we take the limit as $T \to +\infty$ and apply the *Lebesgue's Monotone Convergence* theorem, yielding:

$$\sum_{t=1}^{+\infty} \mathbb{E}\left[ \mathbb{I}_{[f(\theta_t) - f^* > a_1]} \epsilon_t \|\nabla f(\theta_t)\|^2 \right] < C(p, C_\eta, C_p).$$

Due to $a_1 < C_p$, we have

$$\sum_{t=1}^{+\infty} \mathbb{E}\left[\mathbb{I}_{[f(\theta_t)-f^* \geq C_p]}\epsilon_t \|\nabla f(\theta_t)\|^2\right] < C(p, C_\eta, C_p).$$

With this, we complete the proof.

$\square$

## C.8 THE PROOF OF LEMMA 4.1

*Proof.* It is easy to see that, for any $T \geq 1$, and $M := \max\{f(\theta_1) - f^*, C_p\}$, we have

$$\mathbb{E}\left[\sup_{1 \leq t < T} (f(\theta_t) - f^*)\right] \leq \mathbb{E}\left[\mathbb{I}_{\sup_{1 \leq t < T}(f(\theta_t)-f^*) \leq M} \sup_{1 \leq t < T} (f(\theta_t) - f^*)\right]$$

$$+ \mathbb{E}\left[\mathbb{I}_{\sup_{1 \leq t < T}(f(\theta_t)-f^*) > M} \sup_{1 \leq t < T} (f(\theta_t) - f^*)\right]$$

$$\leq M + \underbrace{\mathbb{E}\left[\mathbb{I}_{\sup_{1 \leq t < T}(f(\theta_t)-f^*) > M} \sup_{1 \leq t < T} (f(\theta_t) - f^*)\right]}_{\Omega_T}. \qquad (36)$$

Next, we focus on the $\Omega_T$ on the right-hand side of the above inequality, specifically considering the case where $\sup_{1 \leq t < T}(f(\theta_t) - f^*) > M$. Since we restrict the interval to be between 1 and $T$, it follows that the supremum can indeed be attained, and it is simply the maximum value. Let $\theta_{t^*}$ denote the point where this maximum is attained. Moreover, since $M$ is chosen to be greater than the initial value $f(\theta_1) - f^*$, we can certainly find $t^{**} := \max\{t \leq t^* : f(\theta_t) - f^* \leq M\}$. We can then proceed with the following derivation:

$$\Omega_T = \mathbb{E}\left[\mathbb{I}_{\sup_{1 \leq t < T}(f(\theta_t)-f^*) > M} \sup_{1 \leq t < T} (f(\theta_t) - f^*)\right]$$

$$= \mathbb{E}\left[\mathbb{I}_{\sup_{1 \leq t < T}(f(\theta_t)-f^*) > M} (f(\theta_{t^*}) - f^*)\right]$$

$$\leq \mathbb{E}\left[\mathbb{I}_{\sup_{1 \leq t < T}(f(\theta_t)-f^*) > M} (f(\theta_{t^{**}}) - f^*)\right]$$

$$+ \underbrace{\mathbb{E}\left[\mathbb{I}_{[\sup_{1 \leq t < T}(f(\theta_t)-f^*) > M] \cap [f(\theta_{t^{**}})-f^* < C_p]} \sum_{t=t^{**}}^{t^*-1} |f(\theta_{t+1}) - f(\theta_t)|\right]}_{\Omega_{T,1}}$$

$$+ \underbrace{\mathbb{E}\left[\mathbb{I}_{[\sup_{1 \leq t < T}(f(\theta_t)-f^*) > M] \cap [f(\theta_{t^{**}})-f^* \geq C_p]} \sum_{t=t^{**}}^{t^*-1} |f(\theta_{t+1}) - f(\theta_t)|\right]}_{\Omega_{T,2}}$$

$$\leq M + \Omega_{T,1} + \Omega_{T,2}. \qquad (37)$$

Next, we first compute $|f(\theta_{t+1}) - f(\theta_t)|$, and we obtain:

$$|f(\theta_{t+1}) - f(\theta_t)| \overset{Lagrange's\ Mean\ Value\ theorem}{=} |\nabla f(\theta_{\zeta_t})^\top (\theta_{t+1} - \theta_t)|$$

$$= |\nabla f(\theta_t)^\top (\theta_{t+1} - \theta_t) + (\nabla f(\theta_{\zeta_t}) - \nabla f(\theta_t))^\top (\theta_{t+1} - \theta_t)|$$

$$\leq \|\nabla f(\theta_t)\| \|\theta_{t+1} - \theta_t\| + \|\nabla f(\theta_{\zeta_t}) - \nabla f(\theta_t)\| \|\theta_{t+1} - \theta_t\|$$

$$\overset{L\text{-Smooth}}{\leq} \|\nabla f(\theta_t)\| \|\theta_{t+1} - \theta_t\| + L\|\theta_{t+1} - \theta_t\|^2$$

$$= \epsilon_t \|\nabla f(\theta_t)\| \|g_t\| + L\|g_t\|^2,$$

where $\theta_{\zeta_t}$ is a point between $\theta_{t+1}$ and $\theta_t$. Next, we can compute $\Omega_{T,1}$ and $\Omega_{T,2}$. For $\Omega_{T,1}$, we have:

$$\Omega_{T,1} \leq 1 + \mathbb{E}\left[\mathbb{I}_{[\sup_{1 \leq t < T}(f(\theta_t)-f^*) > M] \cap [f(\theta_{t^{**}})-f^* < C_p]} \overline{\Delta}_{t^{**},1}\right]$$

$$+ \mathbb{E}\left[\mathbb{I}_{[\sup_{1 \le t < T}(f(\theta_t)-f^*)>M] \cap [f(\theta_{t^{**}})-f^*<C_p]} \sum_{t=t^{**}+1}^{t^*-1} |f(\theta_{t+1}) - f(\theta_t)|\right]$$

$$\le 1 + \sum_{t=1}^{T} \mathbb{E}[\overline{\Delta}_t] + \sum_{t=1}^{T} \mathbb{E}\left[\mathbb{I}_{f(\theta_t)-f^* \ge C_p} |f(\theta_{t+1}) - f(\theta_t)|\right]$$

$$\le 1 + \sum_{t=1}^{T} \mathbb{E}[\overline{\Delta}_t] + \left(\sqrt{G}\left(1+\frac{1}{\eta}\right) + L\epsilon_1 G\left(1+\frac{1}{\eta^2}\right)\right) \sum_{t=1}^{T} \mathbb{E}\left[\mathbb{I}_{f(\theta_t)-f^* \ge C_p} \epsilon_t \|\nabla f(\theta_t)\|^2\right]$$

$$\le 1 + C(p,1) + \left(\sqrt{G}\left(1+\frac{1}{\eta}\right) + L\epsilon_1 G\left(1+\frac{1}{\eta^2}\right)\right) C(p, C_\eta, C_p).$$

The definitions of $C(p,1)$ and $C(p, C_\eta, C_p)$ can be found in Lemma B.5 and Lemma 4.2, respectively. As for $\Omega_{T,2}$, we have:

$$\Omega_{T,2} \le \sum_{t=1}^{T} \mathbb{E}\left[\mathbb{I}_{f(\theta_t)-f^* \ge C_p} |f(\theta_{t+1}) - f(\theta_t)|\right]$$

$$\le \left(\sqrt{G}\left(1+\frac{1}{\eta}\right) + L\epsilon_1 G\left(1+\frac{1}{\eta^2}\right)\right) \sum_{t=1}^{T} \mathbb{E}\left[\mathbb{I}_{f(\theta_t)-f^* \ge C_p} \epsilon_t \|\nabla f(\theta_t)\|^2\right]$$

$$\le \left(\sqrt{G}\left(1+\frac{1}{\eta}\right) + L\epsilon_1 G\left(1+\frac{1}{\eta^2}\right)\right) C(p, C_\eta, C_p).$$

The definitions of $C(p, C_\eta, C_p)$ can be found in Lemma 4.2. Substituting the above estimates for $\Omega_{T,1}$ and $\Omega_{T,2}$ back into Eq. 37, we obtain the estimate for $\Omega_T$. Then, substituting $\Omega_T$ back into Eq. 36, we obtain:

$$\mathbb{E}\left[\sup_{1 \le t < T}(f(\theta_t) - f^*)\right] \le 1 + 2M + C(p,1) + 2\left(\sqrt{G}\left(1+\frac{1}{\eta}\right) + L\epsilon_1 G\left(1+\frac{1}{\eta^2}\right)\right) C(p, C_\eta, C_p).$$

It can be seen that the right-hand side of the above inequality is a constant independent of $T$, and $\sup_{1 \le t < T}(f(\theta_t)-f^*)$ is a monotonically increasing sequence. Therefore, by the *Lebesgue's Monotone Convergence* theorem, we obtain:

$$\mathbb{E}\left[\sup_{t \ge 1}(f(\theta_t) - f^*)\right] \le 1 + 2M + C(p,1) + 2\left(\sqrt{G}\left(1+\frac{1}{\eta}\right) + L\epsilon_1 G\left(1+\frac{1}{\eta^2}\right)\right) C(p, C_\eta, C_p)$$

$$< +\infty.$$

With this, we complete the proof.

$\square$

