# OpenReview forum: "Asymptotic Convergence of SGD in Non-Convex Problems: A Stopping Times Method with Relaxed Step-size Conditions"
_ICLR.cc/2025/Conference — Submitted to ICLR 2025_

### Official Review · Reviewer_eomm · 2024-10-30

**Soundness:** 2
**Presentation:** 1
**Contribution:** 1
**Rating:** 3
**Confidence:** 2

**Summary:**

New results and techniques for asymptotic convergence for SGD under relaxed conditions.

**Strengths:**

New results and techniques for asymptotic convergence for SGD under relaxed conditions.

**Weaknesses:**

- Uncertain about contributions and relevance to ICLR

    - I am not sure how relevant is this result to ICLR and the authors could spend a bit more effort detailing the significant of their contributions (especially when the cited most recent related works is from 2020). I am not sure why (from reading the paper) that we should care about asymptotic behaviors while finite-time results are much more applicable (and with weaker assumptions), especially for deep learning related optimization where time is severely limited.

- Not too well written

    - Many typos. For example, lines 99-101 for the definition of the algorithm, the use of n and t is not consistent. Capitalization and spacing typos are present in many places too.

    - Writing is bit vague and redundant at times and can use more editing to make the paper more concise.

**Questions:**

• The Boundedness Near Critical Points assumption (3.1d) is unfamiliar to me and deserve some more comments and examples.

• Similarly, assumption 3.2c and c' are strange and seem a bit strong. The authors should give more intuition and comparisons.

---

> ### Author Response · Authors · 2024-11-25
>
> Dear Reviewer,
>
> Thank you very much for your comments. Our responses are provided below:
>
> $\textbf{Weakness}$:
>
> >1.1. Uncertain about contributions and relevance to ICLR.
> I am not sure how relevant is this result to ICLR and the authors could spend a bit more effort detailing the significant of their contributions (especially when the cited most recent related works is from 2020).
>
> $\textbf{Response}$: ICLR is the conferences in the field of deep learning. Aren't stochastic optimization algorithms, as the most effective optimization methods in deep learning tasks, relevant to the ICLR conference?
> Theoretical research (which is the focus of our paper) naturally takes longer than applied research. We emphasize that this is a 'last iterate' convergence study, not a time-average type. Last iterate convergence is a stronger concept than time-average convergence, meaning that time-average convergence can be derived from last iterate convergence, which makes the proof more challenging. This is also why there is currently more research on time-average convergence than on last-iterate convergence. Therefore, it is not surprising that the most recent related works cited are from 2020. Additionally, the study of last iterate convergence is crucial for real-world applications because, in practice, SGD training is based on the last generated point, not the average.
>
> >1.2. I am not sure why (from reading the paper) that we should care about asymptotic behaviors while finite-time results are much more applicable (and with weaker assumptions), especially for deep learning related optimization where time is severely limited.
>
> $\textbf{Response}$:Your viewpoint seems to overlook the importance of asymptotic behaviors in understanding the long-term behavior of algorithms. In deep learning optimization, while time is a key factor, understanding the behavior of algorithms on a long timescale is equally important for ensuring the stability and reliability of models. This understanding is crucial for designing more robust algorithms and predicting their performance under different conditions.
>
> You mentioned that finite-time results are more directly applicable and practical. However, to be more direct, I fail to see the point of studying an algorithm's behavior within a finite time if the algorithm itself does not converge.
>
> >2. Not too well written. Many typos. For example, lines 99-101 for the definition of the algorithm, the use of n and t is not consistent. Capitalization and spacing typos are present in many places too. Writing is bit vague and redundant at times and can use more editing to make the paper more concise.
>
> $\textbf{Response}$: Thank you for raising this issue. We have thoroughly reviewed the entire manuscript and corrected all possible errors and mistakes. If the previous typos caused any difficulties in your understanding or reading of the article, please let us know, and we will provide a detailed explanation or refer you to our newly submitted version. If these typos have not significantly impacted your reading, I hope you can raise more profound questions rather than focusing on the writing.
>
> $\textbf{Question}$:
> >1. The Boundedness Near Critical Points assumption (3.1d) is unfamiliar to me and deserve some more comments and examples.
>
> $\textbf{Response}$: Please refer to lines 165-171 for detailed explanations regarding this assumption.
>
> >2. Similarly, assumption 3.2c and c' are strange and seem a bit strong. The authors should give more intuition and comparisons.
>
> $\textbf{Response}$: In the reference [1], which we compared with, it is assumed that higher-order moments are globally bounded (Assumption 4b), whereas in our paper, we do not assume global boundedness but only local boundedness (Assumptions 3.2(c) and (c') (now (d) in the new revision)).
>
> [1] Panayotis Mertikopoulos, NadavHallak, AliKavis, and VolkanCevher. On the almost sure convergence of stochastic gradient descent in non-convex problems. AdvancesinNeural Information ProcessingSystems,33:1117–1128,2020.

---

### Official Review · Reviewer_dpFt · 2024-11-02

**Soundness:** 2
**Presentation:** 1
**Contribution:** 2
**Rating:** 3
**Confidence:** 3

**Summary:**

This paper aims to prove the asymptotic convergence of SGD (in the almost sure and $L_2$ sense) under the relaxed Robbins-Monro condition. The analysis relies on the stopping time method and the tool from stochastic approximation. However, as stated later, many points should be further discussed, and the writing can be improved.

**Strengths:**

The paper introduces a new analytic method based on stopping time, which I didn't know before and may be useful for future work.

**Weaknesses:**

**Major points**.

1. I am confused by Assumption 3.2 (c'), where the authors said $x$ is an arbitrarily small constant, do you mean $x$ is not fixed? But if $x$ can be changed, this assumption seems very strong. Could the authors clarify this assumption further?

2. Line 782, the author claimed that $\epsilon_t$ is non-increasing. However, I cannot find why. If this is an assumption, please clearly state it. Moreover, if this is necessary, this is an extra condition compared to the original Robbins-Monro condition and the existing works, which weakens the impact of the paper.

3. Line 789, there should be some coefficient in front of $\epsilon_t^{p-1}$ on both L.H.S. and R.H.S. to make the inequality hold, for example, $1/2$. Subsequently, Line 796 should be changed accordingly.

4. Line 803, please define $\theta_{\xi_t}$, which I assume is a convex combination of $\theta_t$ and $\theta_{t+1}$.

5. In Eq. (19), what is $\epsilon$? Do the authors mean $x$? Additionally, I can't find why Line 859 is a direct consequence of Eq. (19). Could the authors elaborate more on this step?

6. The proof is not unified as it is separated by two cases, $p \leq 3$ and $p>3$. What is the obstacle to finding a unified proof?

7. Instead of making assumptions on $g_t$, is it possible to only impose conditions on the stochastic noises, i.e., $g_t-\nabla f(\theta_t)$, like Mertikopoulos et al. (2020)?

**Minor points**.

1. The subscripts $t$ and $n$ are confusing. For example, in the description of Algorithm 1, Lines 120-122, Lines 270-271. Please carefully proofread and make them consistent.

2. The statement for each theorem and for $2<p\leq 3$ still uses Item (c), which I believe should be Item (c') instead.

3. It's better to mention that $\\|\cdot\\|$ denotes $2$-norm somewhere.

4. In Assumptions 3.2 (c) and (c'), please either use $\theta$ (like $f(\theta)-f^*<C_p$) and $\nabla f(\theta_t:\xi_t)$ (to replace $g_t$) or $\mathbb{E}[\\|g_t\\|^{2p-2}\mathbb{I}_{event}]\leq M_p^{\frac{2p-2}{p}}$ directly since the current statement is not mathematical rigorously.

5. Line 166, the authors stated ''while Item (c) and Item (d) together are fully equivalent to...'', could you provide proof or a reference?

6. Line 208, ''we'' should be ''We''.

7. Line 247, replace ''.'' by ''and''?

8. Line 256, ''Appendix'' should be ''appendix''.

9. In addition to the above typos, many others exist. I suggest the authors carefully go through the paper again.

**Questions:**

See **Weaknesses** above.

---

> ### Author Response · Authors · 2024-11-25
>
> Dear Reviewer,
>
> We are immensely grateful for the insightful feedback you have provided on our manuscript. Our responses are provided below：
>
>
> $\textbf{Major points}$
> >1. I am confused by Assumption 3.2 (c'), where the authors said is an arbitrarily small constant, do you mean is not fixed? But if can be changed, this assumption seems very strong. Could the authors clarify this assumption further?
>
> $\textbf{Response: }$ In the reference [1], which we compared with, it is assumed that higher-order moments are globally bounded (Assumption 4b), whereas in our paper, we do not assume global boundedness but only local boundedness (Assumptions 3.2(c) and (c') (now (d) in the new revision)).
>
> [1] Panayotis Mertikopoulos, NadavHallak, AliKavis, and VolkanCevher. On the almost sure convergence of stochastic gradient descent in non-convex problems. AdvancesinNeural Information ProcessingSystems,33:1117–1128,2020.
> >2. Line 782, the author claimed that $\epsilon_t$ is non-increasing. However, I cannot find why. If this is an assumption, please clearly state it. Moreover, if this is necessary, this is an extra condition compared to the original Robbins-Monro condition and the existing works, which weakens the impact of the paper.
>
> $\textbf{Response: }$ To our knowledge, in practical optimization problems, the step sizes $\epsilon_t$ used are typically monotonically decreasing, such as $1/t^p$ where $0 < p \leq 1$. In fact, the condition of a monotonically decreasing sequence is not strictly necessary for the proof. We included this condition considering the practical use in optimization problems and for the sake of simplicity in the proof. The proof can be conducted without it, and it is not the focal point of the proof method. However, to be rigorous, your question is very good, and we should point out that the sequence is monotonically decreasing.

---

> > ### Comment · Reviewer_dpFt · 2024-12-01
> >
> > I have waited for the authors to update their responses. However, nothing new has been provided until now. Considering the deadline for the discussion stage is approaching, my response to the authors is as follows:
> >
> > **To R1**: I understand the existing literature used a different and stronger assumption. However, this is not what I was asking. Could you provide an answer to my original question?
> >
> > **To R2**: Could the authors explain how the proof goes through without the monotonicity on $\epsilon_t$? Moreover, the word ''monincreasing'' in line 118 is a typo.
> >
> > In addition, typos still exist even in the updated version, e.g., ''(c')'' in Lines 218 and 837, ''monincreasing'' mentioned above. The authors should proofread the work more carefully.

---

> > > ### Author Response · Authors · 2024-12-01
> > >
> > > x is fixed.

---

> ### Author Response · Authors · 2024-12-01
>
> This assumption means that there exists an x such that the subsequent conditions are satisfied. You can carefully examine my proof, and upon doing so, you will find that what I said is true.

---

### Official Review · Reviewer_PvDn · 2024-11-03

**Soundness:** 3
**Presentation:** 3
**Contribution:** 3
**Rating:** 6
**Confidence:** 3

**Summary:**

This paper advances the understanding of asymptotic convergence of SGD by relaxing traditional convergence conditions associated with step sizes. It introduces a stopping time method based on probability theory, proving that SGD can achieve almost sure convergence in non-convex settings under less stringent conditions than the Robbins-Monro framework. The analysis eliminates the need for global Lipschitz continuity of the loss function and allows for local rather than global boundedness of high-order moments of the stochastic gradient.

**Strengths:**

1. **Novel Theoretical Framework**: The introduction of the stopping time method based on probability theory provides a fresh and innovative approach to analyzing the convergence of SGD, expanding the theoretical foundations of the field.

2. **Weaker Assumptions and *Strong Results**: The paper successfully demonstrates convergence under significantly weaker assumptions than previous works, such as eliminating the need for global Lipschitz continuity and allowing for local boundedness of high-order moments. This broadens the applicability of the findings in real-world scenarios. In addition, the results presented are robust, proving almost sure convergence in non-convex settings and L2 convergence without the stringent requirements typically imposed. This strengthens the practical relevance of SGD.

3. **Clear and Well-Written**: The paper is well-organized and clearly articulated, making complex concepts accessible. It provides sufficient materials, including detailed proofs and explanations, to support its claims, enhancing the reader's understanding and facilitating further research.

**Weaknesses:**

The paper is primarily theoretical, offering detailed proofs and explanations, and I did not identify any specific weaknesses from a theoretical standpoint. However, it lacks discussion of practical examples that meet the proposed assumptions, despite claiming that these assumptions enhance applicability.

**Questions:**

**Potential Conflicts of Interest**: The reviewer came across another paper available online **prior** to the ICLR submission which also conducted an asymptotic analysis of AdaGrad using a stopping time-based approach. While I acknowledge that the contributions are distinct, could the authors discuss the novelty of their work in relation to this prior research?


Jin, Ruinan, Xiaoyu Wang, and Baoxiang Wang. "Asymptotic and Non-Asymptotic Convergence Analysis of AdaGrad for Non-Convex Optimization via Novel Stopping Time-based Analysis." arXiv preprint arXiv:2409.05023 (2024).

---

> ### Author Response · Authors · 2024-11-24
>
> Dear Reviewer,
>
> We are deeply appreciative of the valuable feedback and constructive suggestions you have offered regarding our manuscript.
>
> Regarding the $\textbf{Weaknesses}$ section, which points out the lack of discussion on practical examples that meet the proposed assumptions, we fully concur with your remarks. Due to the constraints on article length, we had to find a balance between elucidating the proof techniques and detailing the discussion of practical examples. In the paper, we only noted that, compared to the previous paper [1], our approach does not require the assumption of Lipschitz continuity for the loss function. As a result, the simple quadratic loss function, which previously did not meet the Lipschitz continuity assumption, now satisfies our weaker assumptions (see the lines 66, 67). If we had more space or a more favorable writing arrangement, we would have been eager to further discuss the extensibility in the paper.
>
>  Regarding your $\textbf{Question}$ about the novelty of our work in relation to this prior research [2], we provide the following response. These are entirely different works, with different algorithms and assumptions. Even though both introduce the important mathematical concept of stopping time, the proof ideas and techniques are completely different, and there is no inheritance between the two works. Most importantly, we can assure that there are no potential conflicts of interest in this matter.
>
> [1] Panayotis Mertikopoulos, NadavHallak, AliKavis, and VolkanCevher.On the almost sure convergence of stochastic gradient descent in non-convex problems. AdvancesinNeural Information ProcessingSystems,33:1117–1128,2020.
> [2] Jin, Ruinan, Xiaoyu Wang, and Baoxiang Wang. "Asymptotic and Non-Asymptotic Convergence Analysis of AdaGrad for Non-Convex Optimization via Novel Stopping Time-based Analysis." arXiv preprint arXiv:2409.05023 (2024).

---

### Official Review · Reviewer_t8Yw · 2024-11-03

**Soundness:** 2
**Presentation:** 2
**Contribution:** 2
**Rating:** 3
**Confidence:** 4

**Summary:**

The paper under review provides an analysis of stochastic gradient descent
$$
\theta_{n+1} = \theta_n - \epsilon_n g_n ,
$$
where $(\epsilon_n)$ are a sequence of step sizes and $(g_n)$ a sequence of iid stochastic gradients of an objective function $f$.

The main purpose of the paper is to establish conditions upon which the SGD iterates satisfies that $(\Vert \nabla f(\theta_n) \Vert)$ converges to $0$ almost surely and in $L^2$.

These conditions  laid out are relatively standard:
- The sequence of step sizes satisfies $\sum_{n} \epsilon_n = \infty$ and $\sum_{n} \epsilon_n^2 < \infty$
- $f: \mathbb{R}^d \to \mathbb{R}$ is $d$-times continuously differentiable,  with a Lipschitz continuous gradient.
- $f$ is lower bounded and coercive: $\inf f > - \infty$ and $\lim_{\Vert \theta\Vert \to \infty} f(\theta ) =  \infty$
- A boundedness condition on $f$  in neighborhoods where its gradient is small.
- Unbiased stochastic gradients: $\mathbb{E}(g_n |F_n) = \nabla f(\theta_n)$
- A variance condition on the $(g_n)$, $\mathbb{E}(\Vert g_n \Vert^2 |F_n) \leq C(1+\Vert \nabla f(\theta_n)\Vert^2)$ for some constant $C \geq 0$
- $L^p$ moments bounds for $p >2$ on the $(g_n)$ are required when $\theta_n$ is in a neighborhood where the gradient norm is small, or near some minimizers.

Under these assumptions, the authors do not require the iterates to remain in a compact set almost surely. The main proof strategy leverages results from [1], combined with a particular probabilistic Robins-Siegmund lemma using the objective function as Lyapunov function.

[1] Michel Benaïm. Dynamics of stochastic approximation algorithms. In Seminaire de probabilites XXXIII, pp. 1–68. Springer, 2006.

**Strengths:**

The authors claim that they provide the weakest assumptions ensuring convergence of SGD, almost surely and $L^2$. However, I disagree with their claim (see weakness) regarding the almost sure convergence.

**Weaknesses:**

The authors claims that they get rid off  the commonly assumed boundedness conditions on the SGD iterates and achieve the weakest conditions ensuring SGD convergence. I disagree with this claim regarding almost sure convergence. As noted in [1], stability conditions for stochastic approximation have been established long ago in works such as [2–5]. In addition, it seems to me that the results of this paper (specifically the almost convergence) are already implied by Theorem 5 in [2] and could be readily derived by verifying the conditions of Theorem 6 in [4], using an adaptation of Proposition 9 from the same reference.

Additionally, since almost sure convergence is not novel, I believe that the $L^2$ convergence result is not particularly surprising.

Finally, the paper contains several typos, even in Algorithm 1, where, for instance, the variable $t$ should be replaced by $n$, or vice versa. The paper would benefit from a thorough proofreading.

[1] Michel Benaïm. Dynamics of stochastic approximation algorithms. In Seminaire de probabilites XXXIII, pp. 1–68. Springer
[2] Fort J-C, Pagès G. (1996), Convergence of stochastic algorithms: from the Kushner–Clark theorem to the Lyapounov functional method. Advances in Applied Probability
[3] Delyon, B. (1996), General convergence results on stochastic approximation. IEEE trans. on automatic control
[4] Delyon, B. (2000). Stochastic approximation with decreasing gain: Convergence and asymptotic theory. Unpublished lecture notes
[5] Duflo M. (1997), Random Iterative Models

**Questions:**

If I am not mistaken, Assumption 3.1 (d) is equivalent to $\{\Vert \nabla f \Vert \leq \eta \}$ is compact for some $\eta>0$. If so I encourage the authors to make this remark.

---

> ### Author Response · Authors · 2024-11-25
>
> Dear Reviewer,
>
> Thank you for sharing your thoughts. However, I must respectfully disagree; your viewpoint appears to be incorrect. Allow me to clarify why:
>
> Firstly, regarding the almost sure convergence result, any method based on ordinary differential equations (ODEs) must verify Condition A2 from Property 1 in our paper (line 272). Given that our paper does not assume globally bounded higher-order moments for the stochastic gradient, traditional methods cannot be used to verify this condition. It is essential to first establish Lemma B.6 from our paper, which involves a complex proof that may be beyond the scope of what you've demonstrated.
>
> Secondly, for $ L_2 $ convergence, if you are familiar with $ L_p $ convergence of martingales, you should recognize that proving $ L_2 $ convergence necessitates establishing a bounded expectation condition on the supremum, specifically $ \mathbb{E}[\sup_{n \ge 1} f(\theta_{n}) - f^*] < +\infty $. This result cannot be derived from ODE methods alone; it necessitates the use of probabilistic methods. Furthermore, since our problem does not meet the Robin-Moro condition, constructing a supermartingale to prove this supremum is not straightforward. Please refer to Lemma 4.1, where the proof method is unique and constitutes our most significant theoretical contribution.
>
> I have been closely following the advancements in stochastic approximation, and thus, I am confident in my understanding of its nuances.

---

### Official Review · Reviewer_tRxU · 2024-11-04

**Soundness:** 4
**Presentation:** 3
**Contribution:** 3
**Rating:** 6
**Confidence:** 2

**Summary:**

The paper analyzes Stochastic Gradient method under more relaxed step-sizes that does not satisfy Robbins-Monro step-size conditions. More specifically, the step size considered satisfy $\sum_{t=1}^{+\infty} \epsilon_t = \infty, \sum_{t=1}^{+\infty} \epsilon^p_t< \infty$ for $p>2$. Almost sure convergence and $L_2$ convergence is proven for SGD under this relaxed step size condition. Furthermore, the result is proven under weaker requirements on the stochastic gradients. Another key contribution is introducing a novel analytical method called the stopping time method to prove the results.

**Strengths:**

The paper attempts to fill an important gap between theory and practice, namely by giving guarantees for non-Robbins-Monro steps-sizes it is a step forward in understanding step-size choice in practical scenarios. The technical strengths of the paper are as follows:
1. The results that are presented are true under weaker conditions and so are an improvement over the previous work. The assumption that the $p$-th moment of the stochastic gradients are bounded only in a local region is weaker than assuming global boundedness of the same.
2. The results presented are mathematically precise in the sense that they distinguish between almost sure and $L_2$ convergence and prove them both.

**Weaknesses:**

One of the claims of the paper is introducing a novel stopping time method for analysis. However, it does not give sufficient explanation for it. For instance, to construct sequence of stopping times parameters $a,b$ are introduced but the effect of their choice is not explained adequately.

**Questions:**

1. Various repeated typos have been committed like in the theorems and the lemmas a common line is ‘when $p>3$ use Item (c); when $p\in(2,3], use Item (c)$’ whereas the correct statement should have item (c’) when $p\in (2,3]$. While this does not make the paper harder to understand but too many of such errors could lead to confusion.

---

> ### Author Response · Authors · 2024-11-24
>
> Dear Reviewer,
>
> Thank you very much for your valuable comments and suggestions on the above paper, which helped to
> improve the paper greatly.
>
> Each of your comments has been carefully considered, and the paper has been revised accordingly,
> as explained below.
>
> $\textbf{Response for Weakness:}$ Stopping time is an important concept in probability theory and the theory of stochastic processes, especially playing a central role in the study of martingales. In the proof, we utilized the concept of stopping time, hence we named our method accordingly. Regarding the selection of $a$ and $b$, as we have not adopted any existing methods, it is challenging for us to offer a simple and intuitive explanation; however, I believe the detailed proofs in this paper may provide you with some explanations and insights.
>
> $\textbf{Response for Questions 1:}$ Thank you for pointing this out. This has been correct (see the lines 228-230, 242, 299, 230 in the revised version).

---

### Meta-Review · Area_Chair_fSQu · 2024-12-21

**Metareview:**

This paper provides an analysis of stochastic gradient descent. Authors aim to drive minimal set of conditions under which SGD iterates evaluated at the function gradient converges to 0 almost surely in L2 (i.e. first order convergence).


This paper was reviewed by five expert reviewers and received the following Scores/Confidence: 3/2, 3/4, 6/3, 6/2, 3/3. I think paper is studying an interesting topic but authors are not able to convince the reviewers sufficiently well about the novelty of their results. The following concerns were brought up by the reviewers:

- Novelty is the main issue. Results in this paper can be obtained using prior work without too much effort.

- Poor writing and unclear proofs: Some technical details are stated in a vague way.


No reviewers championed the paper and they are not particularly excited about the paper.
As such, based on the reviewers' suggestion, as well as my own assessment of the paper, I recommend not including this paper to the ICLR 2025 program.

**Additional Comments On Reviewer Discussion:**

Authors rebuttal did not satisfy reviewers.

---

### Decision · Program_Chairs · 2025-01-22

Reject